# Efficient Methods for Non-stationary Online Learning

**Peng Zhao, Yan-Feng Xie, Lijun Zhang, Zhi-Hua Zhou**
National Key Laboratory for Novel Software Technology,
Nanjing University, Nanjing, China
{zhaop, xieyf, zhanglj, zhouzh}@lamda.nju.edu.cn

## Abstract

Non-stationary online learning has drawn much attention in recent years. In particular, *dynamic regret* and *adaptive regret* are proposed as two principled performance measures for online convex optimization in non-stationary environments. To optimize them, a two-layer online ensemble is usually deployed due to the inherent uncertainty of the non-stationarity, in which a group of base-learners are maintained and a meta-algorithm is employed to track the best one on the fly. However, the two-layer structure raises the concern about the computational complexity — those methods typically maintain $\mathcal{O}(\log T)$ base-learners simultaneously for a $T$-round online game and thus perform multiple projections onto the feasible domain per round, which becomes the computational bottleneck when the domain is complicated. In this paper, we present efficient methods for optimizing dynamic regret and adaptive regret, which reduce the number of projections per round from $\mathcal{O}(\log T)$ to $1$. Moreover, our obtained algorithms require only one gradient query and one function evaluation at each round. Our technique hinges on the reduction mechanism developed in parameter-free online learning and requires non-trivial twists on non-stationary online methods. Empirical studies verify our theoretical findings.

## 1 Introduction

Classic online learning minimizes the static regret, which benchmarks the online learner's performance against the best fixed decision in hindsight. In many real-world online applications, however, the environments are non-stationary [Zhou, 2022] and static regret becomes less attractive since it would be unrealistic to assume the existence of a single decision behaved satisfactorily over time.

To address the limitation, in recent years, researchers have studied more strengthened performance measures to facilitate online algorithms with the capability of handling non-stationarity. In particular, dynamic regret [Zinkevich, 2003; Zhang et al., 2018a] and adaptive regret [Hazan and Seshadhri, 2009; Daniely et al., 2015] are proposed as two principled metrics to guide the algorithm design. We focus on the online convex optimization (OCO) setting [Hazan, 2016]. OCO can be deemed as a game between the learner and the environments. At each round $t \in [T]$, the learner submits her decision $\mathbf{x}_t \in \mathcal{X}$ from a convex feasible domain $\mathcal{X} \subseteq \mathbb{R}^d$ and simultaneously environments choose a convex function $f_t : \mathcal{X} \mapsto \mathbb{R}$, and subsequently the learner suffers an instantaneous loss $f_t(\mathbf{x}_t)$.

### 1.1 Dynamic Regret and Adaptive Regret

Dynamic regret is proposed by Zinkevich [2003] to compare the online learner's performance against a sequence of *any* feasible comparators $\mathbf{u}_1, \ldots, \mathbf{u}_T \in \mathcal{X}$. Formally, it is defined as

$$\text{D-REG}_T(\mathbf{u}_1, \ldots, \mathbf{u}_T) = \sum_{t=1}^{T} f_t(\mathbf{x}_t) - \sum_{t=1}^{T} f_t(\mathbf{u}_t). \tag{1}$$

36th Conference on Neural Information Processing Systems (NeurIPS 2022).

Dynamic regret minimization enables the learner to track changing comparators. A favorable dynamic regret bound should scale with a certain non-stationarity measure dependent on the comparators such as the path length $P_T = \sum_{t=2}^{T} \|\mathbf{u}_t - \mathbf{u}_{t-1}\|_2$. Notably, the classic static regret can be treated as a special case of dynamic regret by specifying the comparators as the best fixed decision in hindsight.

Adaptive regret is proposed by Hazan and Seshadhri [2009] and further strengthened by Daniely et al. [2015], which measures the regret over *any* interval $I = [r, s] \subseteq [T]$ with a length of $\tau = |I|$, i.e.,

$$\text{A-REG}_T(|I|) = \max_{[r, r+\tau-1] \subseteq [T]} \left\{ \sum_{t=r}^{r+\tau-1} f_t(\mathbf{x}_t) - \min_{\mathbf{u} \in \mathcal{X}} \sum_{t=r}^{r+\tau-1} f_t(\mathbf{u}) \right\}. \tag{2}$$

Since the minimizers of different intervals can be different, adaptive regret minimization also ensures the capability of competing with changing comparators. A desired adaptive regret bound should be as close as the minimax static regret of this interval. Algorithms with adaptive regret matching static regret of this interval up to logarithmic terms in $T$ are referred to strongly adaptive [Daniely et al., 2015]. Moreover, adaptive regret includes static regret when choosing $I = [T]$.

It is worth noting that the relationship between dynamic regret and adaptive regret for OCO is generally unclear [Zhang, 2020, Section 5], even though a black-box reduction from dynamic regret to adaptive regret has been proven for the simper expert setting (i.e., online linear optimization over simplex) [Luo and Schapire, 2015, Theorem 4]. Hence, the two measures are separately developed and many algorithms have been proposed, including algorithms for dynamic regret [Zinkevich, 2003; Hall and Willett, 2013; Zhang et al., 2018a; Zhao et al., 2020, 2021b,a; Baby and Wang, 2021; Zhao et al., 2022a] and the ones for adaptive regret [Hazan and Seshadhri, 2009; Daniely et al., 2015; Jun et al., 2017; Zhang et al., 2018b, 2019]. Note that there are also studies [Zhang et al., 2020; Cutkosky, 2020] optimizing both measures simultaneously by an even strengthened metric $\sum_{t=r}^{s} f_t(\mathbf{x}_t) - \sum_{t=r}^{s} f_t(\mathbf{u}_t)$ over any interval $[r, s] \subseteq [T]$, hence called "interval dynamic regret".

## 1.2 Two-layer Structure and Projection Complexity Issue

The fundamental challenge of optimizing these two non-stationary regret measures is the uncertainty of the environmental non-stationarity. Concretely, to ensure the robustness to the unknown environments, dynamic regret aims to compete with *any* feasible comparator sequence, while adaptive regret examines the local performance over *any* intervals. The unknown comparators or unknown intervals bring considerable uncertainty to online optimization. To address the issue, a two-layer structure is usually deployed to optimize the measures, where a set of base-learners are maintained to handle the different possibilities of online environments and a meta-algorithm is employed to combine them all and track the unknown best one. Such a framework successfully achieves many state-of-the-art results, including the $\mathcal{O}(\sqrt{T(1 + P_T)})$ dynamic regret [Zhang et al., 2018a] and the $\mathcal{O}(\sqrt{(F_T + P_T)(1 + P_T)})$ small-loss dynamic regret for smooth functions [Zhao et al., 2020], where $P_T = \sum_{t=2}^{T} \|\mathbf{u}_t - \mathbf{u}_{t-1}\|_2$ is the path length and $F_T = \sum_{t=1}^{T} f_t(\mathbf{u}_t)$ is the cumulative loss of comparators; as well as the $\mathcal{O}(\sqrt{|I| \log T})$ adaptive regret [Jun et al., 2017] and the $\mathcal{O}(\sqrt{F_I \log F_I \log F_T})$ small-loss adaptive regret for smooth functions [Zhang et al., 2019] for any interval $I = [r, s] \subseteq [T]$, where $F_I = \min_{\mathbf{x} \in \mathcal{X}} \sum_{t=r}^{s} f_t(\mathbf{x})$ and $F_T = \min_{\mathbf{x} \in \mathcal{X}} \sum_{t=1}^{T} f_t(\mathbf{x})$. Besides, an $\mathcal{O}(\sqrt{|I|(\log T + P_I)})$ interval dynamic regret is also achieved by a two-layer (or even three-layer) structure [Zhang et al., 2020], where $P_I = \sum_{t=r}^{s} \|\mathbf{u}_t - \mathbf{u}_{t-1}\|_2$ is the path length over the interval.

The two-layer methods have demonstrated great effectiveness in tackling non-stationary online environments, whereas the gain is at the price of heavier computations than the methods for minimizing static regret. While it is believed that additional computations are necessary for more robustness, we are wondering whether it is possible to pay for a "minimal" computation overhead for adapting to the non-stationarity. To this end, we focus on the popular first-order online methods and aim to streamline unnecessary computations while retaining the same regret guarantees. Arguably, the most computationally expensive step of each round is the projection onto the convex feasible domain, namely, the projection operation $\Pi_{\mathcal{X}}[\mathbf{y}] = \arg\min_{\mathbf{x} \in \mathcal{X}} \|\mathbf{x} - \mathbf{y}\|_2$ for a convex set $\mathcal{X} \subseteq \mathbb{R}^d$. Typical two-layer non-stationary online algorithms require maintaining $N = \mathcal{O}(\log T)$ base-learners simultaneously to cover the possibility of unknown environments. Define the *projection complexity* of online methods as the number of projections onto the feasible domain per round. Then, those non-stationary methods suffer an $\mathcal{O}(\log T)$ projection complexity, whereas standard online methods for static regret minimization require only one projection per round such as online gradient descent [Zinkevich, 2003].

## 1.3 Our Contributions and Techniques

In this paper, we design a generic mechanism to reduce the projection complexity of many existing non-stationary methods from $\mathcal{O}(\log T)$ to $1$ *without sacrificing the regret optimality*, hence matching the projection complexity of stationary methods. Our reduction is inspired by the recent advance in parameter-free online learning [Cutkosky and Orabona, 2018; Mhammedi et al., 2019]. The idea is simple: we reduce the original problem learned in the feasible domain $\mathcal{X}$ to an alternative one learned in a *surrogate domain* $\mathcal{Y} \supseteq \mathcal{X}$ such that the projection onto it is much cheaper, e.g., simply choosing $\mathcal{Y}$ as a properly scaled Euclidean ball; and moreover, a carefully designed *surrogate loss* is necessary for the alternative problem to retain the regret optimality. We reveal that a necessary condition for our reduction mechanism to deploy and reduce the projection complexity is that the non-stationary online algorithm shall *query the function gradient once and evaluate the function value once per round*. Several algorithms for the worst-case dynamic regret or adaptive regret already satisfy the requirements, so we can immediately deploy the reduction and obtain their efficient counterparts with the same regret guarantees and $1$ projection complexity. However, many non-stationary algorithms, particularly those designed for small-loss bounds, do not satisfy the requirement. Hence, we require non-trivial efforts to make them compatible. Due to this, we have developed a series of algorithms that achieve worst-case/small-loss dynamic regret and adaptive regret with one projection per round (actually, with one gradient query and one function evaluation per round as well).

Despite that the reduction mechanism of this paper has been studied in parameter-free online learning, applying it to non-stationary online learning requires new ideas and non-trivial modifications. Here we highlight the technical innovation. The main challenge comes from the reduction condition mentioned earlier — as the surrogate loss involves the projection operation, our reduction requires the algorithm query one gradient and evaluate one function value at each round. However, many non-stationary algorithms do not satisfy the requirement, which is to be contrasted to the parameter-free algorithms such as MetaGrad [van Erven and Koolen, 2016; Mhammedi et al., 2019] that naturally satisfy the condition. For example, the SACS algorithm [Zhang et al., 2019] enjoys the best known small-loss adaptive regret, yet the method requires $N$ gradient queries and $N+1$ function evaluations at each round, where $N = \mathcal{O}(\log T)$ is the number of base-learners. Thus, we have to dig into the algorithm and modify it to fit our reduction. First, we replace their meta-algorithm with Adapt-ML-Prod [Gaillard et al., 2014], an expert-tracking algorithm with a *second-order* regret with excess losses to accommodate the linearized loss that is used to ensure one gradient query per round. Second, we introduce a sequence of *time-varying* thresholds to adaptively determine the problem-dependent geometric covers in contrast to a fixed threshold used in their method. In particular, we register the cumulative loss of the final decisions rather than the base-learner's one to compare it with the changing thresholds, which renders the design of one function value evaluation per round and also turns out to be crucial for achieving an improved small-loss bound that can recover the best known worst-case adaptive adaptive regret (by contrast, SACS cannot obtain optimal worst-case adaptive regret). To summarize, our final algorithm only requires one projection/gradient query/function evaluation at each round, substantially improving the efficiency of SACS algorithm that requires $N$ projections/gradient queries/function evaluations per round.

## 1.4 Assumptions

We list several standard assumptions used in OCO [Shalev-Shwartz, 2012; Hazan, 2016]. Notably, not all assumptions are always required. We will explicitly state the requirements in the theorem.

**Assumption 1** (bounded gradient)**.** The norm of the gradients of online functions over the domain $\mathcal{X}$ is bounded by $G$, i.e., $\|\nabla f_t(\mathbf{x})\|_2 \leq G$, for all $\mathbf{x} \in \mathcal{X}$ and $t \in [T]$.

**Assumption 2** (bounded domain)**.** The domain $\mathcal{X} \subseteq \mathbb{R}^d$ contains the origin $\mathbf{0}$, and the diameter of the domain $\mathcal{X}$ is at most $D$, i.e., $\|\mathbf{x} - \mathbf{x}'\|_2 \leq D$ for any $\mathbf{x}, \mathbf{x}' \in \mathcal{X}$.

**Assumption 3** (non-negativity and smoothness)**.** All the online functions are non-negative and $L$-smooth, i.e., for any $\mathbf{x}, \mathbf{x}' \in \mathcal{X}$ and $t \in [T]$, $\|\nabla f_t(\mathbf{x}) - \nabla f_t(\mathbf{x}')\|_2 \leq L\|\mathbf{x} - \mathbf{x}'\|_2$.

**Organization.** The rest is structured as follows. Section 2 presents the reduction mechanism and illustrates its application to dynamic regret minimization. Section 3 provides efficient methods for optimizing adaptive regret. Section 4 reports the experiments. Section 5 concludes the paper and makes discussions. All the proofs and omitted details for algorithms are deferred to the appendices.

## 2 The Reduction Mechanism and Dynamic Regret Minimization

We start from the dynamic regret minimization. First, we briefly review existing methods in Section 2.1, and then present our reduction mechanism and illustrate how to apply it to reducing the projection complexity of dynamic regret methods in Section 2.2.

### 2.1 A Brief Review of Dynamic Regret Minimization

Zhang et al. [2018a] propose a two-layer online algorithm called Ader with an $\mathcal{O}(\sqrt{T(1 + P_T)})$ dynamic regret, which is proven to be minimax optimal for convex functions. Ader maintains a group of base-learners, each performing online gradient descent (OGD) [Zinkevich, 2003] with a customized step size specified by the pool $\mathcal{H} = \{\eta_1, \ldots, \eta_N\}$, and then uses a meta-algorithm to combine them all. Denoted by $\mathcal{B}_1, \ldots, \mathcal{B}_N$ the $N$ base-learners. For each $i \in [N]$, $\mathcal{B}_i$ updates by

$$\mathbf{x}_{t+1,i} = \Pi_{\mathcal{X}}[\mathbf{x}_{t,i} - \eta_i \nabla f_t(\mathbf{x}_t)], \tag{3}$$

where $\eta_i \in \mathcal{H}$ is the associated step size and $\Pi_{\mathcal{X}}[\cdot]$ denotes the projection onto the feasible domain $\mathcal{X}$ with $\Pi_{\mathcal{X}}[\mathbf{y}] = \arg\min_{\mathbf{x} \in \mathcal{X}} \|\mathbf{y} - \mathbf{x}\|_2$. Notably, all the base-learners share the same gradient $\nabla f_t(\mathbf{x}_t)$ rather than using their individual one $\nabla f_t(\mathbf{x}_{t,i})$. This is because Ader optimizes the linearized loss $\ell_t(\mathbf{x}) = \langle \nabla f_t(\mathbf{x}_t), \mathbf{x} \rangle$, which enjoys the benign property of $\nabla \ell_t(\mathbf{x}_{t,i}) = \nabla f_t(\mathbf{x}_t)$ for all $i \in [N]$.

Furthermore, the meta-algorithm evaluates each base-learner by $\ell_t(\mathbf{x}_{t,i}) = \langle \nabla f_t(\mathbf{x}_t), \mathbf{x}_{t,i} \rangle$ and updates the weight vector $\boldsymbol{p}_{t+1} \in \Delta_N$ by the Hedge algorithm [Freund and Schapire, 1997], namely,

$$p_{t+1,i} = \frac{p_{t,i} \exp(-\varepsilon \langle \nabla f_t(\mathbf{x}_t), \mathbf{x}_{t,i} \rangle)}{\sum_{j=1}^{N} p_{t,j} \exp(-\varepsilon \langle \nabla f_t(\mathbf{x}_t), \mathbf{x}_{t,j} \rangle)}, \quad \forall i \in [N], \tag{4}$$

where $\varepsilon > 0$ is the learning rate of the meta-algorithm. The final prediction is obtained by $\mathbf{x}_{t+1} = \sum_{i=1}^{N} p_{t+1,i} \mathbf{x}_{t+1,i}$. The learner submits the prediction $\mathbf{x}_{t+1}$ and then receives the loss $f_{t+1}(\mathbf{x}_{t+1})$ and the gradient $\nabla f_{t+1}(\mathbf{x}_{t+1})$ as the feedback of this round. Under a suitable configuration of the step size pool $\mathcal{H}$ with $N = \mathcal{O}(\log T)$ and learning rate $\varepsilon = \Theta(\sqrt{(\ln N)/T})$, Ader enjoys an $\mathcal{O}(\sqrt{T(1 + P_T)})$ dynamic regret [Zhang et al., 2018a, Theorem 4].

For convex and smooth functions, Zhao et al. [2021b] demonstrate that a similar two-layer structure can attain an $\mathcal{O}(\sqrt{(F_T + P_T)(1 + P_T)})$ small-loss dynamic regret under a suitable setting of the step size pool $\mathcal{H}$ and time-varying learning rates of meta-algorithm $\{\varepsilon_t\}_{t=1}^{T}$, where $F_T = \sum_{t=1}^{T} f_t(\mathbf{u}_t)$ is the cumulative loss of the comparators. This bound safeguards the minimax rate in the worst case, while can be much smaller than $\mathcal{O}(\sqrt{T(1 + P_T)})$ bound in benign environments.

### 2.2 The Reduction Mechanism for Reducing Projection Complexity

As demonstrated in the update (3), all the base-learners require projecting the intermediate solution onto the domain $\mathcal{X}$ to ensure the feasibility. As a result, $\mathcal{O}(\log T)$ projections are required at each round, which is generally time-consuming particularly when the domain $\mathcal{X}$ is complicated.

We present a generic reduction mechanism for reducing the projection complexity and apply it to dynamic regret methods. Our reduction builds upon the seminal work [Cutkosky and Orabona, 2018] and a further refined result [Cutkosky, 2020], who propose a black-box reduction from constrained online learning to the unconstrained setting (or another constrained problem with a larger domain) .

**Reduction mechanism.** Given an algorithm for non-stationary online learning Algo whose projection complexity is $\mathcal{O}(\log T)$, our reduction mechanism builds on it to yield an algorithm Efficient-Algo with 1 projection onto $\mathcal{X}$ per round and retaining the same order of regret. The central idea is to replace expensive projections onto the original domain $\mathcal{X}$ with other much cheaper projections. To this end, we introduce a *surrogate domain* $\mathcal{Y}$ defined as the minimum Euclidean ball containing the feasible domain $\mathcal{X}$, i.e., $\mathcal{Y} = \{\mathbf{x} \mid \|\mathbf{x}\|_2 \leq D\} \supseteq \mathcal{X}$. Then, the reduced algorithm Algo works on $\mathcal{Y}$ whose projection can be realized by a simple rescaling. More importantly, to avoid regret degeneration, it is necessary to carefully construct the surrogate loss $g_t : \mathcal{Y} \mapsto \mathbb{R}$ as

$$g_t(\mathbf{y}) = \langle \nabla f_t(\mathbf{x}_t), \mathbf{y} \rangle - \mathbb{1}_{\{\langle \nabla f_t(\mathbf{x}_t), \mathbf{v}_t \rangle < 0\}} \cdot \langle \nabla f_t(\mathbf{x}_t), \mathbf{v}_t \rangle \cdot S_{\mathcal{X}}(\mathbf{y}), \tag{5}$$

---

**Algorithm 1** Efficient Algorithm for Minimizing Dynamic Regret

---

**Input:** step size pool $\mathcal{H} = \{\eta_1, \ldots, \eta_N\}$, learning rate of meta-algorithm $\varepsilon_t$ (or simply a fixed $\varepsilon$).

1: Initialization: let $\mathbf{x}_1$ and $\{\mathbf{y}_{1,i}\}_{i=1}^N$ be any point in $\mathcal{X}$; $\forall i \in [N], p_{1,i} = 1/N$.
2: **for** $t = 1$ **to** $T$ **do**
3:    Receive the gradient information $\nabla f_t(\mathbf{x}_t)$.
4:    Construct the surrogate loss $g_t : \mathcal{Y} \mapsto \mathbb{R}$ according to Eq. (5).
5:    Compute the gradient $\nabla g_t(\mathbf{y}_t)$ according to Lemma 1.
6:    For each $i \in [N]$, the base-learner $\mathcal{B}_i$ produces the local decision by

$$\widehat{\mathbf{y}}_{t+1,i} = \mathbf{y}_{t,i} - \eta_i \nabla g_t(\mathbf{y}_t), \;\; \mathbf{y}_{t+1,i} = \widehat{\mathbf{y}}_{t+1,i}\Big(\mathbb{1}_{\{\|\widehat{\mathbf{y}}_{t+1,i}\|_2 \leq D\}} + \frac{D}{\|\widehat{\mathbf{y}}_{t+1,i}\|_2} \cdot \mathbb{1}_{\{\|\widehat{\mathbf{y}}_{t+1,i}\|_2 \geq D\}}\Big).$$

7:    Meta-algorithm updates weight by $p_{t+1,i} \propto \exp(-\varepsilon_{t+1} \sum_{s=1}^t \langle \nabla g_s(\mathbf{y}_s), \mathbf{y}_{s,i} \rangle), i \in [N]$.
8:    Compute $\mathbf{y}_{t+1} = \sum_{i=1}^N p_{t+1,i}\mathbf{y}_{t+1,i}$.
9:    Submit $\mathbf{x}_{t+1} = \Pi_{\mathcal{X}}[\mathbf{y}_{t+1}]$.      ▷ The only step projects onto feasible domain $\mathcal{X}$ per round.
10: **end for**

---

where $S_{\mathcal{X}}(\mathbf{y}) = \inf_{\mathbf{x} \in \mathcal{X}} \|\mathbf{y} - \mathbf{x}\|_2$ is the distance function to $\mathcal{X}$ and $\mathbf{v}_t = {(\mathbf{y}_t - \mathbf{x}_t)}/{\|\mathbf{y}_t - \mathbf{x}_t\|_2}$ is the vector indicating the projection direction.

The main protocol of our reduction is presented as follows. The input includes original functions $\{f_t\}_{t=1}^T$, the feasible domain $\mathcal{X}$, and the reduced algorithm Algo.

1: **for** $t = 1, \ldots, T$ **do**
2:    receive the gradient information $\nabla f_t(\mathbf{x}_t)$;
3:    construct the surrogate loss $g_t : \mathcal{Y} \mapsto \mathbb{R}$ according to Eq. (5);
4:    obtain the intermediate prediction $\mathbf{y}_{t+1} \leftarrow \mathsf{Algo}(g_t(\cdot), \mathbf{y}_t, \mathcal{Y})$;
5:    submit the final prediction $\mathbf{x}_{t+1} = \Pi_{\mathcal{X}}[\mathbf{y}_{t+1}]$;
6: **end for**

Our reduction enjoys the regret safeness due to the following benign properties of surrogate loss.

**Theorem 1** (Theorem 2 of Cutkosky [2020]). *The surrogate loss $g_t : \mathcal{Y} \mapsto \mathbb{R}$ defined in (5) is convex. Moreover, we have $\|\nabla g_t(\mathbf{y}_t)\|_2 \leq \|\nabla f_t(\mathbf{x}_t)\|_2$ and for any $\mathbf{u}_t \in \mathcal{X}$*

$$\langle \nabla f_t(\mathbf{x}_t), \mathbf{x}_t - \mathbf{u}_t \rangle \leq g_t(\mathbf{y}_t) - g_t(\mathbf{u}_t) \leq \langle \nabla g_t(\mathbf{y}_t), \mathbf{y}_t - \mathbf{u}_t \rangle. \tag{6}$$

The theorem shows the convexity of the surrogate loss $g_t(\mathbf{y})$ and we thus have $f_t(\mathbf{x}_t) - f_t(\mathbf{u}_t) \leq \langle \nabla g_t(\mathbf{y}_t), \mathbf{y}_t - \mathbf{u}_t \rangle$, which implies that it suffices to optimize the linearized upper bound, i.e., to optimize function $\ell_t(\mathbf{y}) = \langle \nabla g_t(\mathbf{y}_t), \mathbf{y} \rangle$. The following lemma specifies the gradient calculation.

**Lemma 1.** *For any $\mathbf{y} \in \mathcal{Y}$, $\nabla g_t(\mathbf{y}) = \nabla f_t(\mathbf{x}_t)$ when $\langle \nabla f_t(\mathbf{x}_t), \mathbf{v}_t \rangle \geq 0$; and $\nabla g_t(\mathbf{y}) = \nabla f_t(\mathbf{x}_t) - \langle \nabla f_t(\mathbf{x}_t), \mathbf{v}_t \rangle \cdot (\mathbf{y} - \Pi_{\mathcal{X}}[\mathbf{y}])/\|\mathbf{y} - \Pi_{\mathcal{X}}[\mathbf{y}]\|_2$ when $\langle \nabla f_t(\mathbf{x}_t), \mathbf{v}_t \rangle < 0$. Here $\mathbf{v}_t = (\mathbf{y}_t - \mathbf{x}_t)/\|\mathbf{y}_t - \mathbf{x}_t\|_2$. In particular, $\nabla g_t(\mathbf{y}_t) = \nabla f_t(\mathbf{x}_t) - \langle \nabla f_t(\mathbf{x}_t), \mathbf{v}_t \rangle \cdot \mathbf{v}_t$ when $\langle \nabla f_t(\mathbf{x}_t), \mathbf{v}_t \rangle < 0$.*

**Reduction requirements.** An important necessary condition for the reduction is to require the reduced algorithm satisfying *one gradient query* and *one function evaluation* at each round. Indeed, the reduction essentially updates according to the surrogate loss $\{g_t\}_{t=1}^T$. Note that the definition of surrogate loss involves the distance function $S_{\mathcal{X}}(\mathbf{y})$, see Eq. (5). Thus, each evaluation of $g_t(\mathbf{y})$ leads to one projection onto $\mathcal{X}$ due to the calculation of $S_{\mathcal{X}}(\mathbf{y})$. Similarly, each gradient query of $\nabla g_t(\mathbf{y})$ also contributes to one projection, see Lemma 1 for details. To summarize, we can use the reduction to ensure a 1 projection complexity, only when the reduced algorithm satisfies the requirements of one gradient query and one function evaluation per round. Below, we demonstrate the usage of our reduction mechanism for two methods of dynamic regret minimization that satisfy the conditions, including the worst-case method [Zhang et al., 2018a] and the small-loss method [Zhao et al., 2021b].

**Application to dynamic regret minimization.** Algorithm 1 summarizes the main procedures of our efficient methods for optimizing dynamic regret, which is an instance of the reduction mechanism by picking Algo as Ader [Zhang et al., 2018a]. More specifically, Lines 6 – 8 are essentially performing Ader algorithm using the surrogate loss $\{g_t\}_{t=1}^T$ over the surrogate domain $\mathcal{Y}$. Note that the base update in Line 6 is essentially performing OGD with projection onto $\mathcal{Y}$, a scaled Euclidean ball, and

thus the projection admits a simple closed form. The overall algorithm requires projecting onto $\mathcal{X}$ only once per round, see Line 9. Our method provably retains the same dynamic regret.

**Theorem 2.** *Set the step size pool as $\mathcal{H} = \left\{ \eta_i = 2^{i-1}(D/G)\sqrt{5/(2T)} \mid i \in [N] \right\}$ with $N = \lceil 2^{-1} \log_2(1 + 2T/5) \rceil + 1$ and the learning rate as $\varepsilon = \sqrt{(\ln N)/(1 + G^2 D^2 T)}$. Under Assumptions 1 and 2, our algorithm requires one projection onto $\mathcal{X}$ per round and enjoys*

$$\sum_{t=1}^{T} f_t(\mathbf{x}_t) - \sum_{t=1}^{T} f_t(\mathbf{u}_t) \leq \mathcal{O}\big(\sqrt{T(1 + P_T)}\big). \tag{7}$$

For smooth and non-negative functions, the Sword++ algorithm [Zhao et al., 2021b] achieves an $\mathcal{O}(\sqrt{(F_T + P_T)(1 + P_T)})$ small-loss dynamic regret, which requires one gradient and one function value per iteration.[1] However, notice that the surrogate loss $g_t(\cdot)$ in Eq. (5) is neither smooth nor non-negative, which hinders the application of our reduction to their method. Fortunately, owing to the benign property of $\|\nabla g_t(\mathbf{y}_t)\|_2 \leq \|\nabla f_t(\mathbf{x}_t)\|_2$ (see Theorem 1), we can still deploy the reduction via an improved analysis and obtain a projection-efficient algorithm with the same small-loss bound.

**Theorem 3.** *Set the step size pool as $\mathcal{H} = \left\{ \eta_i = 2^{i-1}\sqrt{5D^2/(1 + 8LGDT)} \mid i \in [N] \right\}$ with $N = \lceil 2^{-1} \log_2((5D^2 + 2D^2T)(1 + 8LGDT)/(5D^2)) \rceil + 1$ and the learning rate of the meta-algorithm as $\varepsilon_t = \sqrt{(\ln N)/(1 + D^2 \sum_{s=1}^{t-1} \|\nabla g_s(\mathbf{y}_s)\|_2^2)}$. Under Assumptions 1, 2, and 3, our algorithm requires one projection onto $\mathcal{X}$ per round and enjoys the following dynamic regret:*

$$\sum_{t=1}^{T} f_t(\mathbf{x}_t) - \sum_{t=1}^{T} f_t(\mathbf{u}_t) \leq \mathcal{O}\big(\sqrt{(F_T + P_T)(1 + P_T)}\big), \tag{8}$$

*where $F_T = \sum_{t=1}^{T} f_t(\mathbf{u}_t)$ is the cumulative loss of the comparators.*

## 3 Adaptive Regret Minimization

In this section, we present our efficient methods to minimize adaptive regret. First, we briefly review existing methods in Section 3.1, and then present our efficent methods to reducing the projection complexity of adaptive regret methods in Section 3.2.

### 3.1 A Brief Review of Adaptive Regret Minimization

Adaptive regret minimization ensures the online learner to be competitive with a fixed decision over every contiguous interval. For the worst-case bound, the best known result is the $\mathcal{O}(\sqrt{|I|\log T})$ adaptive regret bound achieved by the CBCE algorithm [Jun et al., 2017]. CBCE algorithm requires multiple gradients at each round. Wang et al. [2018] improve CBCE by using the linearized loss to make it requiring one gradient per iteration and retaining the same adaptive regret. Moreover, the improved CBCE algorithm only evaluates the function value once per iteration. Therefore, we can directly apply our reduction and obtain a projection-efficient variation with the same adaptive regret. More detailed elaborations can be found in Appendix C.1.

Now, we focus on the more challenging case of small-loss adaptive regret. The best known result is the $\mathcal{O}(\sqrt{F_I \log F_I \log F_T})$ bound for any interval $I = [r, s] \subseteq [T]$ obtained by the SACS algorithm [Zhang et al., 2019], where $F_I = \min_{\mathbf{x} \in \mathcal{X}} \sum_{t=r}^{s} f_t(\mathbf{x})$ and $F_T = \min_{\mathbf{x} \in \mathcal{X}} \sum_{t=1}^{T} f_t(\mathbf{x})$. However, SACS does not satisfy our reduction requirements, because it requires $N$ gradient queries (i.e., $\nabla f_t(\mathbf{x}_{t,i})$ for $i \in [N]$) and $N + 1$ function evaluations (i.e., $f_t(\mathbf{x}_{t,i})$ for $i \in [N]$, and $f_t(\mathbf{x}_t)$) at round $t \in [T]$, where $N$ denotes the number of active base-learners and $\mathbf{x}_{t,i}$ denotes local decision returned by the $i$-th base-learner. To address so, we have to modify the algorithm to fit our purpose.

In the following, we first sketch the SACS algorithm and then present our modifications. In fact, to optimize the adaptive regret, an online algorithm usually consists of the three components:

---

[1]Sword++ algorithm is mainly proposed for gradient-variation dynamic regret, so there are advanced components (such as correction term and optimism) in algorithm design. It can be verified that their algorithm can be simplified by dropping the correction term and optimism when only small-loss bound is desired.

(i) base-algorithm: an online algorithm that can attain low (static) regret in a given interval;

(ii) scheduling: a set of intervals and each one is associated with a base-learner who aims to minimize the static regret over the interval (from starting time to ending time);

(iii) meta-algorithm: a combining algorithm that can track the best base-learner on the fly.

The specific configurations of the SACS algorithm is as follows. First, SACS uses scale-free online gradient descent (SOGD) [Orabona and Pál, 2018] as the base-algorithm, which ensures a small-loss regret in a given interval. Second, SACS employs AdaNormalHedge [Luo and Schapire, 2015] as the meta-algorithm, which supports the sleeping expert setup and also enjoys a small-loss regret. Finally, SACS designs a clever strategy of *problem-dependent* geometric covers to determine the set of intervals such that the number of active base-learners also depends on the small-loss quantity. As a result, SACS can achieve a fully problem-dependent adaptive regret of order $\mathcal{O}(\sqrt{F_I \log F_I \log F_T})$, scaling with the cumulative loss of comparators. However, SACS also suffers from an $\mathcal{O}(\log T)$ projection complexity in the worst case due to a two-layer structure; and moreover, it can be observed that SACS only attains an $\mathcal{O}(\sqrt{|I| \log |I| \log T})$ bound in the worst case, which exhibits an $\sqrt{\log |I|}$ gap compared with the best known result of $\mathcal{O}(\sqrt{|I| \log T})$ [Jun et al., 2017]. Below, we present an efficient algorithm for small-loss adaptive regret, which resolves the above two issues simultaneously.

## 3.2 Efficient Algorithms for Adaptive Regret

As multiple gradient queries and function evaluations are involved in all the three components of SACS, we have to make plenty of modifications to achieve an algorithm with small-loss adaptive regret yet requiring only one gradient query and function evaluation per round. With such an algorithm on hand, we can then deploy our reduction to achieve an efficient method with 1 projection complexity. Below we present the details. By the reduction mechanism, it is noticeable that we only need to consider the input online functions as surrogate loss $\{g_t\}_{t=1}^T$, where $g_t$ is defined in Eq. (5).

**Base-algorithm.** We use SOGD with a *linearized* surrogate loss $\langle \nabla g_t(\mathbf{y}_t), \mathbf{y} \rangle$ over the surrogate domain $\mathcal{Y}$. Denote by $A_t$ the set of active base-learners' indices, then the base-learner $\mathcal{B}_i$ updates by

$$\mathbf{y}_{t+1,i} = \Pi_{\mathcal{Y}}[\mathbf{y}_{t,i} - \eta_{t,i}\nabla g_t(\mathbf{y}_t)], \tag{9}$$

with $\eta_{t,i} = D/\sqrt{(\delta + \sum_{s=\tau_i}^t \|\nabla g_s(\mathbf{y}_s)\|_2^2)}$, where $\tau_i$ denotes the starting time of the base-learner $i \in A_t$. The projection onto $\mathcal{Y}$ can be easily calculated by a simple rescaling if needed. Notably, owing to the convexity of the surrogate loss $g_t$, we can use the *same* gradient $\nabla g_t(\mathbf{y}_t)$ for all the base-learners at each round, ensuring one gradient query of $\nabla f_t(\mathbf{x}_t)$ at each round.

**Geometric Covers.** The covers consist of a set of intervals that specify the alive time of base-learners. To achieve a small-loss adaptive regret, SACS [Zhang et al., 2019] employs a clever covering construction called problem-dependent geometric covers (PGC) — instead of initiating a base-learner at each round $t$ like earlier algorithms [Daniely et al., 2015; Jun et al., 2017], SACS adds a new base-learner only when the cumulative loss exceeds a pre-defined *threshold*. As a result, the number of active base-learners relates to the small-loss quantity such that the overall algorithm achieves a fully problem-dependent adaptive regret. Notably, to determine the threshold, SACS monitors the cumulative loss of the latest base-learner $f_t(\mathbf{x}_{t,i^\dagger})$ with $i^\dagger$ being the latest base-learner's index, but clearly this will introduction an additional function evaluation beyond $f_t(\mathbf{x}_t)$ at each round.

To avoid the limitation, instead of using a fixed threshold to decide the initiations of base-learners, we design a sequence of *time-varying thresholds* to adaptively start a new base-learner according to amount of cumulative loss of *final decisions* (e.g., $f_t(\mathbf{x}_t)$), bypassing the requirement of additional function evaluation. This realizes the condition of one function evaluation per round. Also, the new design of thresholds mechanism is important to ensure that the overall small-loss bound can simultaneously recover the best known worst-case guarantee, which SACS fails to achieve [Zhang et al., 2019]. Let $C_1, C_2, C_3, \ldots$ denote the sequence of thresholds, and they will be determined by a threshold generating function $\mathcal{G}(\cdot) : \mathbb{N} \mapsto \mathbb{R}_+$ that will be specified later. Our problem-dependent geometric covers are set as follows. We initialize the setting by $s_1 = 1$. We set $s_2$ as the round when the cumulative loss of the overall algorithm (namely, $\sum_{s=1}^t f_s(\mathbf{x}_s)$) exceeds the threshold $C_1$ and then initialize a new instance of SOGD starting at this round. The process is repeated until the end of online game. We thus generate a sequence of points $s_1, s_2, \ldots$, referred to as the *markers*. See

---

**Algorithm 2** Efficient Algorithm for Problem-dependent Adaptive Regret

---

**Input:** threshold generating function $\mathcal{G}(\cdot) : \mathbb{N} \mapsto \mathbb{R}_+$.

1: Initialize total intervals $m = 1$, marker $s_1 = 1$, threshold $C_1 = \mathcal{G}(1)$; let $\mathbf{x}_1$ be any point in $\mathcal{X}$.
2: **for** $t = 1$ **to** $T$ **do**
3:     Receive the gradient information $\nabla f_t(\mathbf{x}_t)$.
4:     Construct the surrogate loss $g_t : \mathcal{Y} \mapsto \mathbb{R}$ according to Eq. (5).
5:     Compute the (sub-)gradient $\nabla g_t(\mathbf{y}_t)$ according to Lemma 1.
6:     Compute $L_t = L_{t-1} + f_t(\mathbf{x}_t)$
        `% constructing Problem-dependent Geometric Covers(PGC)`
7:     **if** $L_t > C_m$ **then**
8:         Set $L_t = 0$, remove base-learners $\mathcal{B}_k$ whose ending point $e_k = m + 1$.
9:         Set $m \leftarrow m + 1$, $s_m \leftarrow t$, $C_m = \mathcal{G}(m)$.
10:        Initialize a new base-learner with ending point $e_m = j$ satisfying $[m, j-1] \in \mathcal{C}$, where $\mathcal{C} = \bigcup_{k \in \mathbb{N} \cup \{0\}} \mathcal{C}_k$ and $\mathcal{C}_k = \left\{ [i \cdot 2^k, (i+1) \cdot 2^k - 1] \mid i \text{ is odd} \right\}$ for all $k \in \mathbb{N} \cup \{0\}$.
11:        Set $\gamma_m = \ln(1 + 2m)$, $w_{t,m} = 1$, $\eta_{t,m} = \min\{1/2, \sqrt{\gamma_m}\}$ for the meta-algorithm.
12:     **end if**
13:     Send $\nabla g_t(\mathbf{y}_t)$ to all base-learners and obtain local predictions $\mathbf{y}_{t+1,i}$ for $i \in A_t$.
14:     Meta-algorithm updates weight $\boldsymbol{p}_{t+1} \in \Delta_{|A_{t+1}|}$ according to Eq. (11), Eq. (12), and Eq. (13)
15:     Compute $\mathbf{y}_{t+1} = \sum_{i \in A_{t+1}} p_{t+1,i} \mathbf{y}_{t+1,i}$.
16:     Submit $\mathbf{x}_{t+1} = \Pi_{\mathcal{X}}[\mathbf{y}_{t+1}]$.         ▷ The only projection onto feasible domain $\mathcal{X}$ per round.
17: **end for**

---

the condition in Line 7, registration of markers in Line 9, and the overall updates in Lines 7 − 11 of Algorithm 2. Those markers specify the starting time (and the ending time) of base-learners and thus construct the PGC as

$$\widetilde{\mathcal{C}} = \bigcup_{k \in \mathbb{N} \cup \{0\}} \widetilde{\mathcal{C}}_k, \text{ where } \widetilde{\mathcal{C}}_k = \left\{ [s_{i \cdot 2^k}, s_{(i+1) \cdot 2^k} - 1] \mid i \text{ is odd} \right\} \text{ for all } k \in \mathbb{N} \cup \{0\}. \quad (10)$$

It is noteworthy to emphasize that PGC is constructed by the language of "marker", whose exact time stamp is *unknown* ahead of time but is only determined according to the learner's performance on the fly. Moreover, the notation $\mathcal{C}$ in Lines 10 of Algorithm 2 is defined based on the registered indexes of markers, and there is a one-one correspondence from the interval in $\mathcal{C}$ to that in PGC $\widetilde{\mathcal{C}}$. More concretely, an interval $[i \cdot 2^k, (i+1) \cdot 2^k - 1] \in \mathcal{C}$ will be mapped into the interval $[s_{i \cdot 2^k}, s_{(i+1) \cdot 2^k} - 1]$ in PGC, managing the alive time of base-learners in a geometric manner with respect to the subscripts.

**Meta-algorithm.** SACS uses the AdaNormalHedge [Luo and Schapire, 2015] as the meta-algorithm, however, this is not suitable for our propose. To ensure one projection per iteration, we cannot use multiple function values, i.e., $\{g_t(\mathbf{y}_{t,i})\}_{i=1}^{N}$, for meta-algorithm to evaluate the loss. Instead, we can only use the *linearized* loss value, namely, $\{\langle \nabla g_t(\mathbf{y}_t), \mathbf{y}_{t,i} \rangle\}_{i=1}^{N}$ in the weight update of meta-algorithm. The small-loss regret bound in the meta-algorithm of SACS crucially relies on the original function values, which is unfortunately inaccessible in our case. Technically, when fed with linearized loss, it is hard to establish a *squared* gradient-norm bound and then convert it to the small loss due to the *first-order* regret bound of AdaNormalHedge. Based on this crucial technical observation, we propose to use the Adapt-ML-Prod algorithm [Gaillard et al., 2014] as the meta-algorithm in our method. The key advantage is that it enjoys a *second-order* regret and also supports the sleeping expert setup. Adapt-ML-Prod maintains multiple learning rates $\boldsymbol{\eta}_{t+1}$ and an intermediate weight vector $\mathbf{w}_{t+1}$, which are updated by the following rule. For any active base-learner $i \in A_{t+1}$,

$$\eta_{t+1,i} = \min \left\{ \frac{1}{2}, \sqrt{\frac{\gamma_i}{1 + \sum_{k=s_i}^{t} (\widehat{\ell}_k - \ell_{k,i})^2}} \right\}, \quad w_{t+1,i} = \left( w_{t,i} \big( 1 + \eta_{t,i} (\widehat{\ell}_t - \ell_{t,i}) \big) \right)^{\frac{\eta_{t+1,i}}{\eta_{t,i}}}, \quad (11)$$

where $\gamma_i = \ln(1 + 2i)$ is a certain scaling factor and the feedback loss is constructed as for $i \in A_t$

$$\widehat{\ell}_t = \langle \nabla g_t(\mathbf{y}_t), \mathbf{y}_t \rangle / (2GD), \text{ and } \ell_{t,i} = \langle \nabla g_t(\mathbf{y}_t), \mathbf{y}_{t,i} \rangle / (2GD). \quad (12)$$

The final weight vector $\boldsymbol{p}_{t+1} \in \Delta_{|A_{t+1}|}$ is obtained by

$$p_{t+1,i} = \frac{w_{t+1,i} \cdot \eta_{t+1,i}}{\sum_{j \in A_{t+1}} w_{t+1,j} \cdot \eta_{t+1,j}}. \quad (13)$$

Notably, the meta update only uses one gradient at round $t$, namely, $\nabla g_t(\mathbf{y}_t)$.

Finally, we compute $\mathbf{y}_{t+1} = \sum_{i \in A_{t+1}} p_{t+1,i} \mathbf{y}_{t+1,i}$ as the overall prediction in the surrogate domain $\mathcal{Y}$ and calculate $\mathbf{x}_{t+1} = \Pi_{\mathcal{X}}[\mathbf{y}_{t+1}]$ to ensure the feasibility. This is the only projection onto $\mathcal{X}$ at each round. Algorithm 2 summarizes the main procedures of our efficient methods for small-loss adaptive regret. Albeit with a similar two-layer structure as SACS, our algorithm exhibits salient differences in base-algorithm, meta-algorithm, and geometric covers. As a benefit, we can successfully deploy our reduction mechanism and make the overall algorithm project onto the feasible domain $\mathcal{X}$ once per round, see Line 16. Our method retains the same small-loss adaptive regret as [Zhang et al., 2019].

**Theorem 4.** *Under Assumptions 1–3, setting the threshold generating function $\mathcal{G}(m) = \Theta(\log m)$ whose explicit form is in Eq. (46) of Appendix C, Algorithm 2 requires only one projection onto $\mathcal{X}$ per round and enjoys the small-loss adaptive regret:*

$$\sum_{t=r}^{s} f_t(\mathbf{x}_t) - \sum_{t=r}^{s} f_t(\mathbf{u}) \leq \mathcal{O}\Big( \min \big\{ \sqrt{F_I \log F_I \log F_T}, \sqrt{|I| \log T} \big\} \Big) \tag{14}$$

*for any interval $I = [r, s] \subseteq [T]$, where $F_I = \min_{\mathbf{x} \in \mathcal{X}} \sum_{t=r}^{s} f_t(\mathbf{x})$ and $F_T = \min_{\mathbf{x} \in \mathcal{X}} \sum_{t=1}^{T} f_t(\mathbf{x})$.*

**Remark 1.** Note that the $\mathcal{O}(\sqrt{F_I \log F_I \log F_T})$ small-loss bound of Zhang et al. [2019] becomes $\mathcal{O}(\sqrt{|I| \log|I| \log T})$ in the worst case, looser than the $\mathcal{O}(\sqrt{|I| \log T})$ bound [Jun et al., 2017] by a factor of $\log|I|$. We show that this limitation can be actually avoided by the new design of thresholds mechanism and a refined analysis. More discussions can be found in Appendix C.3. Indeed, our result in (14) can *strictly* match the best known problem-independent result in the worst case.

## 4 Experiment

In this section, we provide empirical studies to evaluate our proposed methods.

**General Setup.** We conduct experiments on the synthetic data. We consider the following online regression problem. Let $T$ denote the number of total rounds. At each round $t \in [T]$ the learner outputs the model parameter $\mathbf{w}_t \in \mathcal{W} \subseteq \mathbb{R}^d$ and simultaneously receives a data sample $(x_t, y_t)$ with $x_t \in \mathcal{X} \subseteq \mathbb{R}^d$ being the feature and $y_t \in \mathbb{R}$ being the corresponding label.[2] The learner can then evaluate her model by the online loss $f_t(\mathbf{w}_t) = \frac{1}{2}(x_t^\top \mathbf{w}_t - y_t)^2$ which uses a square loss to evaluate the difference between the predictive value $x_t^\top \mathbf{w}_t$ and the ground-truth label $y_t$, and then use the feedback information to update the model. In the simulations, we set $T = 20000$, the domain diameter as $D = 6$, and the dimension of the domain as $d = 8$. The feasible domain $\mathcal{W}$ is set as an ellipsoid $\mathcal{W} = \big\{ \mathbf{w} \in \mathbb{R}^d \mid \mathbf{w}^\top \mathbf{E} \mathbf{w} \leq \lambda_{\min}(\mathbf{E}) \cdot (D/2)^2 \big\}$, where $\mathbf{E}$ is a certain diagonal matrix and $\lambda_{\min}(\mathbf{E})$ denotes its minimum eigenvalue. Then, a projection onto $\mathcal{W}$ requires solving a convex programming that is generally expensive. In the experiment, we use `scipy.optimize.NonlinearConstraint` to solve it to perform the projection onto the feasible domain.

To simulate the non-stationary online environments, we control the way to generate the date samples $\{(x_t, y_t)\}_{t=1}^{T}$. Specifically, for $t \in [T]$, the feature $x_t$ is randomly sampled in an Euclidean ball with a diameter $D$ same as the feasible domain of model parameters; and the corresponding label is set as $y_t = x_t^\top \mathbf{w}_t^* + \varepsilon_t$, where $\varepsilon_t$ is the random noise drawn from $[0, 0.1]$ and $\mathbf{w}_t^*$ is the underlying ground-truth model from the feasible domain $\mathcal{W}$ generated according to a certain strategy specified below. For dynamic regret minimization, we simulate *piecewise-stationary* model drifts, as dynamic regret will be linear in $T$ and thus vacuous when the model drift happens every round due to a linear path length measure. Concretely, we split the time horizon evenly into 25 stages and restrict the underlying model parameter $\mathbf{w}_t^*$ to be stationary within a stage. For adaptive regret minimization, we simulate *gradually evolving* model drifts, where the underlying model parameter $\mathbf{w}_{t+1}^*$ is generated based on the last-round model parameter $\mathbf{w}_t^*$ with an additional random walk in the feasible domain $\mathcal{W}$. The step size of random walk is set to be proportional to $D/T$ to ensure a smooth model change.

**Contenders.** For both dynamic regret and adaptive regret minimization, we directly work on the small-loss online methods. We choose the Sword algorithm [Zhao et al., 2021b] as the contender of

---

[2]With a slight abuse of notations, we here use $\mathbf{w}$ to denote the model parameter and $\mathcal{W}$ to denote the feasible domain, while reserve the notations of $x$ and $\mathcal{X}$ to denote the feature and feature space following the conventional notations of machine learning terminologies.

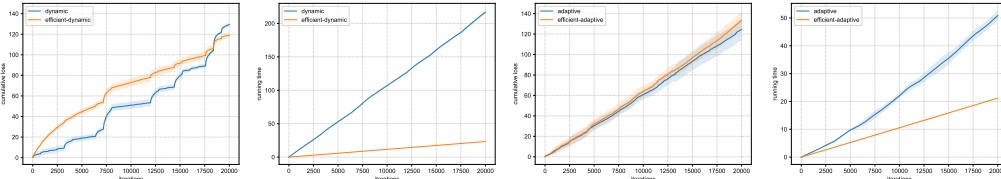

(a) dynamic regret (loss) (b) dynamic regret (time) (c) adaptive regret (loss) (d) adaptive regret (time)

Figure 1: Performance comparisons of existing methods and our methods (indicated by "efficient" prefix) in terms of cumulative loss and running time (in seconds). The first two figures plot the results of methods for dynamic regret minimization, while the latter ones are for adaptive regret.

our efficient method for dynamic regret (Algorithm 1) and choose the SACS algorithm [Zhang et al., 2019] as the contender of our efficient method for adaptive regret (Algorithm 2).

**Results.**   We repeat the experiments for five times with different random seeds and report the results (mean and standard deviation) in Figure 1. We use a machine with a single CPU (Intel(R) Core(TM) i9-10900K CPU @ 3.70GHz) and 32GB main memory to conduct the experiments. We plot both cumulative loss and running time (in seconds) for all the methods. We first examine the performance of dynamic regret minimization, see Figure 1(a) for cumulative loss and see Figure 1(b) for running time. The empirical results show that our method has a comparable performance to Sword without much sacrifice of cumulative loss, while our method can achieve about 10 times speedup due to the improved projection complexity. Second, as shown in Figure 1(c) and Figure 1(d), a similar performance enhancement also appears in adaptive regret minimization, though the speedup is slightly smaller due to the fact that fewer learners are required to maintain for adaptive regret. To summarize, the empirical results show the effectiveness of our methods in retaining the regret performance and also the efficiency in terms of the running time due to the reduced projection complexity.

## 5 Conclusion

In this paper, we design a generic reduction mechanism that can reduce the projection complexity of two-layer methods for non-stationary online learning, hence approaching a clearer resolution of necessary computational overhead for robustness to non-stationarity. Building on the reduction mechanism, we develop a series of online algorithms for optimizing dynamic regret and adaptive regret. All the algorithms retain the best known regret guarantees, and more importantly, require one projection onto the feasible domain per iteration. It is further worth mentioning that, due to the requirement of our reduction, all our algorithms only need one gradient query and one function evaluation at each round as well, which can be appealing in situations with limited feedback.

Our reduction can also be applied to other settings to achieve light project complexity, for example, dynamic regret of OCO with memory [Zhao et al., 2022b], OCO with switching cost [Zhang et al., 2021], and related applications such as online non-stochastic control [Hazan et al., 2020]. Moreover, it is possible to derive similar efficient algorithms for minimizing the interval dynamic regret, an even stringent measure for non-stationary online convex optimization. There is one important open question left on another type of problem-dependent bound that scales with gradient variation [Chiang et al., 2012], which plays an important role in establishing fast convergence in zero-sum games [Syrgkanis et al., 2015; Zhang et al., 2022]. Although Zhao et al. [2021b] have devised a two-layer method that enjoys a gradient-variation dynamic regret and requires one gradient per iteration, it is quite challenging to incorporate the optimistic online learning into our reduction mechanism due to the constrained feasible domain and the complicated two-layer structure. Finally, it would be greatly important to further understand the minimal computational overhead in response to the robustness to non-stationarity, in particular, some information-theoretic arguments would be highly interesting.

## Acknowledgements

This research was supported by NSFC (61921006, 62206125), JiangsuSF (BK20220776), National Postdoctoral Program for Innovative Talent, and Collaborative Innovation Center of Novel Software Technology and Industrialization.

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
