# A  Omitted Details for Reduction Mechanism

In this section, we provide the proofs of Theorem 1 and Lemma 1.

## A.1  Properties of Distance Function

Before presenting the proofs, we here collect two useful lemmas regarding the distance function used in the surrogate loss, which will be useful in the following proofs. The proofs of the two lemmas can be found in the seminal paper of Cutkosky and Orabona [2018].

**Lemma 2** (Proposition 1 of Cutkosky and Orabona [2018]). *The distance function* $S_{\mathcal{X}}(\mathbf{y}) = \inf_{\mathbf{x} \in \mathcal{X}} \|\mathbf{y} - \mathbf{x}\|_2$ *is convex and 1-Lipschitz for any closed convex feasible domain* $\mathcal{X} \subseteq \mathbb{R}^d$.

**Lemma 3** (Theorem 4 of Cutkosky and Orabona [2018]). *Let* $\mathcal{X} \subseteq \mathbb{R}^d$ *a closed convex set. Given* $\mathbf{y} \in \mathbb{R}^d$ *and* $\mathbf{y} \notin \mathcal{X}$. *Let* $\mathbf{x} = \Pi_{\mathcal{X}}[\mathbf{y}]$. *Then we have* $\{\frac{\mathbf{y} - \mathbf{x}}{\|\mathbf{y} - \mathbf{x}\|_2}\} = \partial S_{\mathcal{X}}(\mathbf{y})$.

## A.2  Proof of Theorem 1

Theorem 1 is originally due to Cutkosky [2020], and for self-containedness we restate their proof using our notations.

*Proof.* When $\langle \nabla f_t(\mathbf{x}_t), \mathbf{v}_t \rangle \geq 0$, by the definition of the surrogate loss defined in Eq. (5), we have $g_t(\mathbf{y}) = \langle \nabla f_t(\mathbf{x}_t), \mathbf{y} \rangle$, which is linear in $\mathbf{y}$ and thus convex (in fact linear in $y$). It is clear that $\|\nabla g_t(\mathbf{y}_t)\|_2 = \|\nabla f_t(\mathbf{x}_t)\|_2$ and thus satisfies the claimed inequality of gradient norms in the statement. Moreover, the inequality (6) holds evidently due to the linear surrogate loss in this case.

Let us focus on the case when $\langle \nabla f_t(\mathbf{x}_t), \mathbf{v}_t \rangle < 0$. First, it can be verified that the surrogate loss $g_t(\mathbf{y}) = \langle \nabla f_t(\mathbf{x}_t), \mathbf{y} \rangle - \langle \nabla f_t(\mathbf{x}_t), \mathbf{v}_t \rangle \cdot S_{\mathcal{X}}(\mathbf{y})$ is convex due to the convexity of $S_{\mathcal{X}}(\mathbf{y})$ shown in Lemma 2 and the condition of $\langle \nabla f_t(\mathbf{x}_t), \mathbf{v}_t \rangle < 0$ in this case. Next, the gradient of $g_t(\cdot)$ at the $\mathbf{y}_t$ point can be calculated according to Lemma 1 as,

$$\nabla g_t(\mathbf{y}_t) = \nabla f_t(\mathbf{x}_t) - \langle \nabla f_t(\mathbf{x}_t), \mathbf{v}_t \rangle \cdot \mathbf{v}_t$$

where $\mathbf{v}_t = (\mathbf{y}_t - \mathbf{x}_t)/\|\mathbf{y}_t - \mathbf{x}_t\|_2$. Notice that $\|\mathbf{v}_t\|_2 = 1$ and $\nabla g_t(\mathbf{y}_t)$ is an orthogonal projection of $\nabla f_t(\mathbf{x}_t)$ onto the subspace perpendicular to the vector $\mathbf{v}_t$, so we have $\|\nabla g_t(\mathbf{y}_t)\|_2 \leq \|\nabla f_t(\mathbf{x}_t)\|_2$. Finally, we proceed to prove the inequality (6) in this case. Since the comparator $\mathbf{u}_t \in \mathcal{X}$ is in the feasible domain, we have $S_{\mathcal{X}}(\mathbf{u}_t) = \|\mathbf{u}_t - \mathbf{u}_t\|_2 = 0$ and get

$$
\begin{aligned}
&\langle \nabla f_t(\mathbf{x}_t), \mathbf{x}_t - \mathbf{u}_t \rangle \\
&= \langle \nabla f_t(\mathbf{x}_t), \mathbf{y}_t \rangle + \langle \nabla f_t(\mathbf{x}_t), \mathbf{x}_t - \mathbf{y}_t \rangle - \langle \nabla f_t(\mathbf{x}_t), \mathbf{u}_t \rangle \\
&= \langle \nabla f_t(\mathbf{x}_t), \mathbf{y}_t \rangle - \langle \nabla f_t(\mathbf{x}_t), \frac{\mathbf{y}_t - \mathbf{x}_t}{\|\mathbf{y}_t - \mathbf{x}_t\|_2} \rangle \cdot \|\mathbf{y}_t - \mathbf{x}_t\|_2 - \langle \nabla f_t(\mathbf{x}_t), \mathbf{u}_t \rangle \\
&= \langle \nabla f_t(\mathbf{x}_t), \mathbf{y}_t \rangle - \langle \nabla f_t(\mathbf{x}_t), \mathbf{v}_t \rangle \cdot S_{\mathcal{X}}(\mathbf{y}_t) - \langle \nabla f_t(\mathbf{x}_t), \mathbf{u}_t \rangle + \langle \nabla f_t(\mathbf{x}_t), \mathbf{v}_t \rangle \cdot S_{\mathcal{X}}(\mathbf{u}_t) \\
&= g_t(\mathbf{y}_t) - g_t(\mathbf{u}_t) \\
&\leq \langle \nabla g_t(\mathbf{y}_t), \mathbf{y}_t - \mathbf{u}_t \rangle,
\end{aligned}
$$

where the last inequality holds owing to the convexity of the surrogate loss proven earlier.

Combining the two cases finishes the proof. $\square$

## A.3  Proof of Lemma 1

Lemma 1 is originally due to Cutkosky and Orabona [2018], and for self-containedness we restate their proof using our notations.

*Proof.* With a slight abuse of notations, for simplicity we use the notation $\nabla g_t(\mathbf{y})$ to denote the (sub-)gradient of surrogate function $g_t(\cdot)$ at point $\mathbf{y}$, no matter whether the function is differentiable.

When $\langle \nabla f_t(\mathbf{x}_t), \mathbf{v}_t \rangle \geq 0$, the surrogate loss is $g_t(\mathbf{y}) = \langle \nabla f_t(\mathbf{x}_t), \mathbf{y} \rangle$ by definition in Eq. (5). Therefore, the gradient simply becomes $\nabla g_t(\mathbf{y}_t) = \nabla f_t(\mathbf{x}_t)$.

When $\langle \nabla f_t(\mathbf{x}_t), \mathbf{v}_t \rangle < 0$, the surrogate loss becomes $g_t(\mathbf{y}) = \langle \nabla f_t(\mathbf{x}_t), \mathbf{y} \rangle - \langle \nabla f_t(\mathbf{x}_t), \mathbf{v}_t \rangle \cdot S_{\mathcal{X}}(\mathbf{y})$ according to definition in Eq. (5). By Lemma 3, the gradient $\nabla g_t(\mathbf{y})$ can be calculated by

$$\nabla g_t(\mathbf{y}) = \nabla f_t(\mathbf{x}_t) - \langle \nabla f_t(\mathbf{x}_t), \mathbf{v}_t \rangle \cdot \frac{\mathbf{y} - \Pi_{\mathcal{X}}[\mathbf{y}]}{\|\mathbf{y} - \Pi_{\mathcal{X}}[\mathbf{y}]\|_2},$$

where the computation needs the projection onto domain $\mathcal{X}$. In particular, for $\mathbf{y}_t$, we have

$$\nabla g_t(\mathbf{y}_t) = \nabla f_t(\mathbf{x}_t) - \langle \nabla f_t(\mathbf{x}_t), \mathbf{v}_t \rangle \cdot \frac{\mathbf{y}_t - \mathbf{x}_t}{\|\mathbf{y}_t - \mathbf{x}_t\|_2} = \nabla f_t(\mathbf{x}_t) - \langle \nabla f_t(\mathbf{x}_t), \mathbf{v}_t \rangle \cdot \mathbf{v}_t.$$

This ends the proof. $\qquad\square$

# B  Omitted Details for Dynamic Regret Minimization

In this section, we provide the proofs for the theorems presented in Section 2. Specifically, we first prove the worst-case bound (Theorem 2) and then work on the small-loss bound (Theorem 3).

## B.1  Proof of Theorem 2

*Proof.* Notice that Zhang et al. [2018a] propose the improved Ader algorithm (see Algorithm 3 and Algorithm 4 in their paper), which uses the linearized loss as the input to make the online algorithm requiring one gradient and one function evaluation per iteration. So the algorithm satisfies the requirements of our reduction mechanism, and our algorithm can be regarded as the improved Ader equipped with the projection-efficient reduction. As a consequence, we can directly obtain the same dynamic regret guarantee and ensure 1 projection complexity at the same time, by following the same proof of the improved Ader as well as the reduction guarantee (Theorem 1). $\qquad\square$

## B.2  Proof of Theorem 3

*Proof.* By the reduction guarantee shown in Theorem 1, we have the following result that decomposes the dynamic regret into the two terms.

$$\sum_{t=1}^{T} f_t(\mathbf{x}_t) - \sum_{t=1}^{T} f_t(\mathbf{u}_t) \leq \sum_{t=1}^{T} g_t(\mathbf{y}_t) - \sum_{t=1}^{T} g_t(\mathbf{u}_t) \leq \sum_{t=1}^{T} \langle \nabla g_t(\mathbf{y}_t), \mathbf{y}_t - \mathbf{u}_t \rangle$$

$$= \underbrace{\sum_{t=1}^{T} \langle \nabla g_t(\mathbf{y}_t), \mathbf{y}_t - \mathbf{y}_{t,i} \rangle}_{\texttt{meta-regret}} + \underbrace{\sum_{t=1}^{T} \langle \nabla g_t(\mathbf{y}_t), \mathbf{y}_{t,i} - \mathbf{u}_t \rangle}_{\texttt{base-regret}}, \qquad (15)$$

where in (15) the first term is called *meta-regret* as it measures the regret overhead of the meta-algorithm to track the unknown best base-learner, and the second term is called the *base-regret* to denote the dynamic regret of the base-learner $i$. Note that the above decomposition holds for any base-learner index $i \in [N]$.

**Upper bound of meta-regret.**    As the meta-algorithm can be regarded as a FTRL with time-varying learning rates and a negative entropy regularizer, we apply Lemma 10 to obtain an upper bound for the meta-regret. Indeed, by choosing $\ell_{t,i} = \langle \nabla g_t(\mathbf{y}_t), \mathbf{y}_{t,i} \rangle$ in Lemma 10, we can achieve that

$$\sum_{t=1}^{T} \langle \nabla g_t(\mathbf{y}_t), \mathbf{y}_t - \mathbf{y}_{t,i} \rangle \leq 3 \sqrt{\ln N \left( 1 + \sum_{t=1}^{T} D^2 \|\nabla g_t(\mathbf{y}_t)\|_2^2 \right)} + \frac{G^2 D^2 \sqrt{\ln N}}{2}$$

$$\leq 3D \sqrt{\ln N \sum_{t=1}^{T} \|\nabla g_t(\mathbf{y}_t)\|_2^2} + \frac{(6 + G^2 D^2)\sqrt{\ln N}}{2}$$

$$\leq 3D \sqrt{\ln N \sum_{t=1}^{T} \|\nabla f_t(\mathbf{x}_t)\|_2^2} + \mathcal{O}(1)$$

$$\leq 6D\sqrt{L \ln N \sum_{t=1}^{T} f_t(\mathbf{x}_t)} + \mathcal{O}(1), \tag{16}$$

where the first inequality holds because we have $\|\boldsymbol{\ell}_t\|_\infty^2 = \max_{i \in [N]}(\langle \nabla g_t(\mathbf{y}_t), \mathbf{y}_{t,i} \rangle)^2 \leq D^2 \|\nabla g_t(\mathbf{y}_t)\|_2^2$ by Cauchy-Schwarz inequality and $\|\nabla g_t(\mathbf{y}_t)\|_2 \leq \|\nabla f_t(\mathbf{x}_t)\|_2 \leq G$ (see Theorem 1), the second inequality makes use of $\sqrt{a+b} \leq \sqrt{a} + \sqrt{b}$, the third inequality holds by the property of the surrogate loss (also via Theorem 1), and the last inequality is due to the self-bounding properties of smooth functions (see Lemma 12). Note that $\mathcal{O}(\ln N) = \mathcal{O}(\log \log T)$ can be treated as a constant following previous studies [Luo and Schapire, 2015; Gaillard et al., 2014]

**Upper bound of base-regret.** According to Lemma 7 and noticing that the comparator sequence $\mathbf{u}_1, \ldots, \mathbf{u}_T \in \mathcal{X} \subseteq \mathcal{Y}$ and the diameter of $\mathcal{Y}$ equals to $2D$ by definition, with slight modifications, we have the following dynamic regret bound.

$$\sum_{t=1}^{T} \langle \nabla g_t(\mathbf{y}_t), \mathbf{y}_{t,i} - \mathbf{u}_t \rangle \leq \frac{5D^2}{2\eta_i} + \frac{D}{\eta_i} \sum_{t=2}^{T} \|\mathbf{u}_t - \mathbf{u}_{t-1}\|_2 + \eta_i \sum_{t=1}^{T} \|\nabla g_t(\mathbf{y}_t)\|_2^2$$

$$\leq \frac{5D^2}{2\eta_i} + \frac{D}{\eta_i} \sum_{t=2}^{T} \|\mathbf{u}_t - \mathbf{u}_{t-1}\|_2 + \eta_i \sum_{t=1}^{T} \|\nabla f_t(\mathbf{x}_t)\|_2^2 \tag{17}$$

$$\leq \frac{5D^2}{2\eta_i} + \frac{D}{\eta_i} \sum_{t=2}^{T} \|\mathbf{u}_t - \mathbf{u}_{t-1}\|_2 + 4\eta_i L \sum_{t=1}^{T} f_t(\mathbf{x}_t), \tag{18}$$

where the second inequality is due to the property of the surrogate loss (see Theorem 1) and the last one is due to the self-bounding property of smooth functions (see Lemma 12).

Note that the property of $\|\nabla g_t(\mathbf{y}_t)\|_2 \leq \|\nabla f_t(\mathbf{x}_t)\|_2$ (see Theorem 1) plays an important role in the above analysis. Although the surrogate functions $\{g_t\}_{t=1}^{T}$ are not guaranteed to be smooth and non-negative, we can upper bound its gradient norm by that defined over the original functions $\{g_t\}_{t=1}^{T}$, which are indeed smooth and non-negative. We thus can utilize the self-bounding properties to establish a small-loss bound for the meta-regret and base-regret.

**Upper bound of dynamic regret.** Plugging the above upper bounds of meta-regret and base-regret together, we achieve

$$\sum_{t=1}^{T} f_t(\mathbf{x}_t) - \sum_{t=1}^{T} f_t(\mathbf{u}_t) \leq 6D\sqrt{L \ln N \sum_{t=1}^{T} f_t(\mathbf{x}_t)} + \frac{5D^2 + 2DP_T}{2\eta_i} + 4\eta_i L \sum_{t=1}^{T} f_t(\mathbf{x}_t) + \mathcal{O}(1), \tag{19}$$

which holds for any base-learner's index $i \in [N]$.

Next, we specify the base-algorithm $\mathcal{E}_i$ compared with. Indeed, we aim at choosing the one with step size closest to the (near-)optimal step size $\eta^* = \sqrt{\frac{5D^2 + 2DP_T}{1 + 8LF_T^{\mathbf{x}}}}$, where we denote by $F_T^{\mathbf{x}} = \sum_{t=1}^{T} f_t(\mathbf{x}_t)$ the cumulative loss of the decisions. By Assumption 1 and Assumption 2, we have $F_T^{\mathbf{x}} \in [0, GDT]$ and then the possible minimum optimal and maximum step size are

$$\eta_{\min} = \sqrt{\frac{5D^2}{1 + 8LGDT}}, \text{ and } \eta_{\max} = \sqrt{5D^2 + 2D^2 T}.$$

The construction of step size pool is by discretizing the interval $[\eta_{\min}, \eta_{\max}]$ with intervals with exponentially increasing length. The step size of each base-learner is designed to be monotonically increasing with respect to the index. Consequently, it is evident to verify that there exists an index $i^* \in [N]$ such that

$$\eta_{i^*} \leq \eta^* \leq \eta_{i^*+1} = 2\eta_{i^*}. \tag{20}$$

As the upper bounds of meta-regret and base-regret hold for any compared base-learner, we can choose the index as $i^*$ in particular. Then the second and the third terms in the inequality (19) satisfy

$$\frac{5D^2 + 2DP_T}{2\eta_{i^*}} + 4\eta_{i^*} LF_T^{\mathbf{x}}$$

$$\leq \frac{5D^2 + 2DP_T}{\eta^*} + 4\eta^* LF_T^{\mathbf{x}}$$

$$\leq \sqrt{(5D^2 + 2DP_T)(1 + 8LF_T^{\mathbf{x}})} + \frac{1}{2}\sqrt{(5D^2 + 2DP_T)(1 + 8LF_T^{\mathbf{x}})}$$

$$\leq 3\sqrt{2(5D^2 + 2DP_T)(1 + LF_T^{\mathbf{x}})}. \tag{21}$$

Substituting inequality (21) into inequality (19), we have,

$$\sum_{t=1}^{T} f_t(\mathbf{x}_t) - \sum_{t=1}^{T} f_t(\mathbf{u}_t)$$

$$\leq 6D\sqrt{L \ln N F_T^{\mathbf{x}}} + 3\sqrt{2(5D^2 + 2DP_T)(1 + LF_T^{\mathbf{x}})} + \mathcal{O}(1)$$

$$\leq \left(6D\sqrt{L \ln N} + 3\sqrt{2L(5D^2 + 2DP_T)}\right)\sqrt{F_T^{\mathbf{x}}} + 3\sqrt{2(5D^2 + 2DP_T)} + \mathcal{O}(1)$$

$$\leq \mathcal{O}\left(\sqrt{(1 + P_T)(F_T + \sqrt{P_T} + \mathcal{O}(1))} + P_T + 1\right)$$

$$= \mathcal{O}\left(\sqrt{(F_T + P_T)(1 + P_T)}\right),$$

where the last inequality holds by Lemma 17. Hence, we complete the proof of Theorem 3. □

## C   Omitted Details for Adaptive Regret Minimization

In this section, we present omitted details for minimizing the worst-case and small-loss adaptive regret. First, in Appendix C.1 we describe the efficient algorithm for attaining the worst-case adaptive regret, which provably enjoys the same guarantee as the prior best known work and meanwhile requires one projection only per iteration. Next, we focus on the proof of the main theorem for small-loss adaptive regret, i.e., Theorem 4. To this end, Appendix C.2 provides three key lemmas, Appendix C.3 presents the proof of Theorem C.3, and Appendix C.4 – Appendix C.6 give the proofs of three key lemmas.

### C.1   Results for the Worst-Case Adaptive Regret

In this section, we present results for the worst-case adaptive regret omitted in the main text. The best known result is the $\mathcal{O}(\sqrt{|I| \log T})$ worst-case adaptive regret attained by the CBCE algorithm [Jun et al., 2017], which is achieved by the coin-betting framework with the sleeping expert mechanism. However, CBCE requires multiple gradients at each round. Wang et al. [2018] improve CBCE by using the linearized loss to make the algorithm requiring one gradient per iteration and retaining the same adaptive regret. Moreover, the improved CBCE algorithm only evaluates the function value once per iteration, and the algorithm is presented in [Wang et al., 2018, Algorithm 4 and Algorithm 5] . Therefore, we can simply feed the surrogate loss $g_t$ constructed in (5) to the improved CBCE of Wang et al. [2018], and the obtained algorithm can ensure the same order of adaptive regret and also require only one projection onto the feasible domain per iteration.

### C.2   Key Lemmas

In this part, we present three key lemmas for proving the small-loss adaptive regret, namely, Theorem 4. We then prove Theorem 4 based on these lemmas in Appendix C.3. Finally, we present the proofs for those lemmas in the following several subsections.

The first lemma gives the upper bound of meta-regret of our efficient method for small-loss adaptive regret, which heavily relies on the structure of the problem-dependent geometric covers.

**Lemma 4.** *Under Assumptions 2 and 3, for any interval $I = [i, j] \in \widetilde{\mathcal{C}}$ in the geometric covers defined in Eq. (10) on which we suppose $m$-th base-learner is active, Algorithm 2 ensures*

$$\sum_{\tau=i}^{t} \langle \nabla g_t(\mathbf{y}_t), \mathbf{y}_t - \mathbf{y}_{t,m} \rangle \leq \mathcal{O}\left(\sqrt{\log(m) \sum_{\tau=i}^{t} f_\tau(\mathbf{x}_\tau)}\right),$$

*which holds for any time stamp $t \in [i, j]$.*

Combining above lemma and the analysis of the base-regret upper bound, we can obtain the following adaptive regret for any interval of the problem-dependent geometric covers.

**Lemma 5.** *Under Assumptions 1, 2, and 3, for any interval $[i, j] \in \widetilde{\mathcal{C}}$ in the geometric covers defined in Eq. (10), on which we assume $m$-th expert-algorithm is active, Algorithm 2 ensures*

$$\sum_{\tau=i}^{t} f_\tau(\mathbf{x}_\tau) - \sum_{\tau=i}^{t} f_\tau(\mathbf{u}) \leq \mathcal{O}\left(\sqrt{\log(m) \sum_{\tau=i}^{t} f_\tau(\mathbf{u})}\right),$$

*which holds for any time stamp $t \in [i, j]$ and any comparator $\mathbf{u} \in \mathcal{X}$.*

The above two lemmas rely on the unknown variable of $m$, and the following lemma presents an upper bound for $m$ in terms of the small-loss quantity $F_t$.

**Lemma 6.** *Under Assumptions 1, 2, and 3, for any interval $[i, j] \in \widetilde{\mathcal{C}}$ and any $t \in [i, j]$, the variable $m$ specified in Lemma 4 and Lemma 5 can be bounded by*

$$m \leq \mathcal{O}\left(F_{[1,t]}\right). \tag{22}$$

*This immediately implies that Algorithm 2 ensures*

$$\sum_{\tau=i}^{t} f_\tau(\mathbf{x}_\tau) - \min_{\mathbf{u} \in \mathcal{X}} \sum_{\tau=i}^{t} f_\tau(\mathbf{u}) \leq \mathcal{O}\left(\sqrt{F_{[i,t]} \log F_{[1,t]}}\right),$$

*where $F_{[a,b]} = \min_{\mathbf{u} \in \mathcal{X}} \sum_{\tau=a}^{b} f_\tau(\mathbf{u})$ denotes the cumulative loss of the comparator within the interval $[a, b] \subseteq [T]$.*

## C.3 Proof of Theorem 4

*Proof.* Recall that Theorem 4 exhibits an $\mathcal{O}\left(\min\{\sqrt{F_I \log F_I \log F_T}, \sqrt{|I| \log T}\}\right)$ adaptive regret for any interval $I = [r, s] \subseteq [T]$, where $F_I = \min_{\mathbf{x} \in \mathcal{X}} \sum_{t=r}^{s} f_t(\mathbf{x})$ and $F_T = \min_{\mathbf{x} \in \mathcal{X}} \sum_{t=1}^{T} f_t(\mathbf{x})$. The bound consists of two parts, including a small-loss bound of $\mathcal{O}(\sqrt{F_I \log F_I \log F_T})$ and a worst-case bound of $\mathcal{O}(\sqrt{|I| \log T})$. Below, we present the proofs of the two bounds respectively.

Before showing the proofs, we emphasize again that our result strictly improves the small-loss bound of Zhang et al. [2019], who give an $\mathcal{O}(\sqrt{F_I \log F_I \log F_T})$ bound that becomes $\mathcal{O}(\sqrt{|I| \log|I| \log T})$ in the worst case and thus is looser than the $\mathcal{O}(\sqrt{|I| \log T})$ problem-independent bound [Jun et al., 2017] by a factor of $\log|I|$. Our regret guarantee consists of the small-loss bound and another worst-case bound acting as a safety guarantee for the worst case. Indeed, in the worst-case situation, our bound becomes $\mathcal{O}(\sqrt{|I| \log T})$ and *strictly* match the best known worst-case result [Jun et al., 2017]. We note that our improvement is owing to a refined analysis in the proof as well as our careful algorithm design that only uses one function evaluations to adaptively determine the geometric covers, which is to be contrasted to SACS [Zhang et al., 2019] that uses the latest base-learner's decision to determine the covers.

**Small-loss regret bound.** Let $s_p$ be the smallest marker that larger than $r$, and let $s_q$ be the largest marker that is not large than $s$, then we have

$$s_{p-1} \leq r < s_p, \text{ and } s_q \leq s < s_{q+1}.$$

We bound the regret over the interval $[r, s_p - 1]$ as,

$$\sum_{t=r}^{s_p-1} f_t(\mathbf{x}_t) - \sum_{t=r}^{s_p-1} f_t(\mathbf{u}) \leq \sum_{t=r}^{s_p-1} f_t(\mathbf{x}_t) \leq \sum_{t=s_{p-1}}^{s_p-1} f_t(\mathbf{x}_t) \leq C_{p-1} + GD. \tag{23}$$

The last inequality is because of the construction rule of marker and the fact that $f_t(\mathbf{x}_t) \leq GD$ for any $t \in [T]$ by Assumptions 1 − 3.

By Lemma 11, we can find $v$ consecutive intervals

$$I_1 = [s_{i_1}, s_{i_2} - 1], \ I_2 = [s_{i_2}, s_{i_3} - 1], \ \ldots, I_v = [s_{i_v}, s_{i_{v+1}} - 1] \in \widetilde{\mathcal{C}}, \tag{24}$$

such that

$$i_1 = p, \ i_v \leq q < i_{v+1}, \text{and } v \leq \lceil \log_2(q - p + 2) \rceil.$$

Notice that,

$$q < i_{v+1} \Rightarrow q + 1 \leq i_{v+1} \Rightarrow s_{q+1} - 1 \leq s_{i_{v+1}} - 1 \Rightarrow s \leq s_{i_{v+1}} - 1.$$

For the neat presentation, we define,

$$\alpha(t) = (27GD + 72D^2L)\ln\left(3 + \frac{8}{C_1}\sum_{\tau=1}^{t} f_\tau(\mathbf{u})\right) + 72D^2L\mu^2(t) + 9GD\mu(t) + 6D\sqrt{\delta} + 288D^2L,$$

$$\beta(t) = 4D\sqrt{L}\left(\sqrt{\ln\left(3 + \frac{8}{C_1}\sum_{\tau=1}^{t} f_\tau(\mathbf{u})\right)} + \frac{\mu(t)}{\sqrt{\ln(3 + 8(\sum_{\tau=1}^{t} f_\tau(\mathbf{u}))/C_1)}} + 2\right),$$

where we use the notations $\mu(t) = \ln(1 + (1 + \ln(1 + t))/(2e)) = \mathcal{O}(\log\log t)$ that can be essentially regarded as a constant and $C_1 = \mathcal{G}(1)$, where $\mathcal{G}(\cdot)$ is the threshold generating function defined in (46).

For intervals $I_1$ to $I_v$, by Lemma 5,

$$\sum_{t=s_p}^{s} f_t(\mathbf{x}_t) - \sum_{t=s_p}^{s} f_t(\mathbf{u}) \leq \sum_{k=1}^{v-1}\left(\alpha(s) + \beta(s)\sqrt{F_{I_k}}\right) + \alpha(s) + \beta(s)\sqrt{F_{[s_{i_v}, s]}} \tag{25}$$

$$\leq v\alpha(s) + \beta(s)\sqrt{vF_{[s_p, s]}}$$

$$\leq v\alpha(s) + \beta(s)\sqrt{vF_I}. \tag{26}$$

Combining (23) and (26), the adaptive regret on any interval $i = [r, s]$ will be

$$\sum_{t=r}^{s} f_t(\mathbf{x}_t) - \sum_{t=r}^{s} f_t(\mathbf{u}) \leq v\alpha(s) + \beta(s)\sqrt{vF_I} + C_{p-1} + GD. \tag{27}$$

Next, we show that $v$ and $C_{p-1}$ are of order $\mathcal{O}(\log F_{[r,s]})$ and $\mathcal{O}(\log F_T)$ respectively. By the definition of the time-varying threshold (see the threshold generating function Eq. (46)) and Lemma 6, the threshold can be bounded as,

$$C_{p-1} \leq (54GD + 168D^2L)\ln\left(3 + \frac{8}{C_1}F_{[1,r]}\right) + 168D^2L\mu^2(T) + 18GD\mu(T) + 6D\sqrt{\delta} + 672D^2L,$$

which is of order $\mathcal{O}(\log F_T)$.

With the same argument as (48), it can be shown that between markers $s_p$ and $s_q$, for any $\mathbf{u}' \in \mathcal{X}$,

$$\sum_{t=s_p}^{s_q - 1} f_t(\mathbf{u}') \geq \frac{C_1}{4}(q - p),$$

which suggests

$$q - p \leq \frac{4}{C_1}\sum_{t=s_p}^{s_q - 1} f_t(\mathbf{u}') \leq \frac{4}{C_1}\sum_{t=r}^{s} f_t(\mathbf{u}').$$

We thus have

$$v \leq \lceil \log_2(q - p + 2) \rceil \leq \left\lceil \log_2\left(\frac{4}{C_1}F_{[r,s]} + 2\right)\right\rceil = \mathcal{O}(\log F_{[r,s]}).$$

Combining the upper bounds of $C_{p-1}$ and $v$ as well as the adaptive regret bound in (27) yields

$$\sum_{t=r}^{s} f_t(\mathbf{x}_t) - \sum_{t=r}^{s} f_t(\mathbf{u}) \leq \mathcal{O}(\log F_I \log F_T) + \mathcal{O}(\sqrt{F_I \log F_I \log F_T}) + \mathcal{O}(\log F_T) + \mathcal{O}(1)$$

$$= \mathcal{O}\left(\sqrt{F_I \log F_I \log F_T}\right),$$

where the last step is true as we follow the same convention in [Zhang et al., 2019] to treat the $\log F_I \log F_T$ as the non-leading term. Hence, we finish the proof of the small-loss adaptive regret.

**Worst-case regret bound.** The above proof aims at obtaining small-loss type regret bound, and one of the key steps is to use Cauchy-Schwarz inequality to bound (25), which results in an additional $\mathcal{O}(\sqrt{\log F_{[r,s]}})$ term. Next, we show that actually this extra term can be avoided by the new design thresholds mechanism and thus asymptotically achieve the same worst-case adaptive regret as the best known result [Jun et al., 2017].

From (43) in Lemma 5, we have that for any interval $I = [i, j]$ in problem-dependent covers defined in (10), the adaptive regret is at most

$$\sum_{t=i}^{j} f_t(\mathbf{x}_t) - \sum_{t=i}^{j} f_t(\mathbf{u}) \leq \mathcal{O}\left(\sqrt{\log T \cdot F_{[i,j]}^{\mathbf{x}}} + \log T\right),$$

where we use the notation $F_{[a,b]}^{\mathbf{x}} = \sum_{t=a}^{b} f_t(\mathbf{x}_t)$ to denote the cumulative loss of the returned decisions within the interval $[a, b] \subseteq [T]$. Then, we can use Lemma 6 to upper bound $m \leq \mathcal{O}(T)$ as only the worst-case behavior matters now.

Moreover, for the consecutive intervals defined in (24), we have the following facts:

$$i_{k+1} \leq 2 \cdot i_k, \ \forall k \in [v], \text{ and } |i_{l+1} - i_l| \leq \frac{1}{2}|i_{l+2} - i_{l+1}|, \ \forall l \in [v-1].$$

The first relationship between consecutive foot-indexes of time markers will be used to show that thresholds will not grow too fast during an interval in the geometric covers, which can verified by the construction of cover defined in (10). The second inequality indicates that the times the cumulative loss of algorithm exceeds thresholds in an interval decreases exponentially from $I_v$ to $I_1$, and this can be verified in the proof of [Zhang et al., 2019, Lemma 11].

For the interval $I_k$ with $k \in [v-1]$ in (24), our algorithm's cumulative loss within the interval is upper bounded by

$$\sum_{t=s_{i_k}}^{s_{i_{k+1}}-1} f_t(\mathbf{x}_t) \leq \left(\sum_{a=i_k}^{i_{k+1}-1} C_a\right) + GD|i_{k+1} - i_k| \leq (GD + C_{i_{k+1}-1})|i_{k+1} - i_k|. \quad (28)$$

We then split a given interval $[r, s]$ into three parts to analyze, namely, the consecutive $v - 1$ intervals $I_1$ to $I_{v-1}$, interval $[r, s_p - 1]$, and $[s_{i_v}, s]$, where notably the last two intervals are not fully covered by any interval in geometric covers. For intervals $I_1$ to $I_{v-1}$, we have

$$\sum_{t=s_{i_1}}^{s_{i_v}-1} f_t(\mathbf{x}_t) - \sum_{t=s_{i_1}}^{s_{i_v}-1} f_t(\mathbf{u}) \leq \sum_{a=1}^{v-1} \mathcal{O}\left(\sqrt{\log T \cdot F_{I_a}^{\mathbf{x}}} + \log T\right)$$

$$\leq \sum_{a=1}^{v-1} \mathcal{O}\left(\sqrt{\log T \cdot C_{i_v-1} \cdot |i_{a+1} - i_a|} + \log T\right)$$

$$\leq \sum_{a=1}^{v-1} \mathcal{O}\left(\sqrt{\log T \cdot C_{i_v-1} \cdot \frac{|i_v - i_{v-1}|}{2^{v-1-a}}} + \log T\right)$$

$$\leq \mathcal{O}\left(v \log T + \sqrt{\log T \cdot C_{i_v-1}} \cdot \sum_{b=0}^{+\infty} \sqrt{\frac{|i_v - i_{v-1}|}{2^b}}\right)$$

$$\leq \mathcal{O}\left(v \log T + \sqrt{\log T \cdot C_{i_v-1} \cdot |i_v - i_{v-1}|}\right),$$

where the second inequality is due to the monotonically increasing property of thresholds, the third inequality is by (28), and the last inequality is by the summation of geometric sequence.

By the setting of time-varying thresholds as specified in Eq. (46), we know $C_{i_v-1} = \mathcal{O}(\log(i_v))$.

Furthermore, $|i_v - i_{v-1}|$ represents the number of markers that our algorithm generates during the interval $I_{v-1}$, so it can be upper bounded as

$$|i_v - i_{v-1}| \le \mathcal{O}\left(\frac{GD|I|}{C_{i_{v-1}}}\right) = \mathcal{O}\left(\frac{GD|I|}{\log(i_{v-1})}\right),$$

because the total loss of the algorithm during $|I|$ is at most $GD|I|$ and we use the smallest threshold $C_{i_{v-1}}$ during the interval $I_{v-1}$ to calculate the maximum number of possible markers. Due to the relationship of $2i_{v-1} \ge i_v$ by (28), we then get

$$\sum_{t=s_{i_1}}^{s_{i_v}-1} f_t(\mathbf{x}_t) - \sum_{t=s_{i_1}}^{s_{i_v}-1} f_t(\mathbf{u}) \le \mathcal{O}\left(v \log T + \sqrt{\log T \cdot C_{i_{v-1}} \cdot |i_v - i_{v-1}|}\right)$$

$$\le \mathcal{O}\left(v \log T + \sqrt{\log T \cdot \log(i_v) \cdot \frac{|I|}{\log(i_{v-1})}}\right)$$

$$\le \mathcal{O}\left(v \log T + \sqrt{\log T \cdot |I|(1 + \frac{1}{\log i_{v-1}})}\right)$$

$$\le \mathcal{O}\left(\log |I| \log T + \sqrt{|I| \log T}\right),$$

where the last inequality is due to the fact that $v$ is of order $\mathcal{O}(\log F_{[r,s]}) = \mathcal{O}(\log |I|)$. Remind that the variable $v$ appears in our analysis by Lemma 11, which is independent of the worst-case analysis.

Now we proceed to upper bound the regret over intervals $[r, s_p - 1]$ and $[s_{i_v}, s]$. By a similar analysis used early, we have

$$\sum_{t=r}^{s_p-1} f_t(\mathbf{x}_t) - \sum_{t=r}^{s_p-1} f_t(\mathbf{u}) \le C_{p-1} \le \mathcal{O}(\log T),$$

and

$$\sum_{t=s_{i_v}}^{s} f_t(\mathbf{x}_t) - \sum_{t=s_{i_v}}^{s} f_t(\mathbf{u}) \le \mathcal{O}\left(\log T + \sqrt{F_{[s_{i_v},s]} \log T}\right) \le \mathcal{O}\left(\log T + \sqrt{|I| \log T}\right).$$

Combining everything together achieves

$$\sum_{t=r}^{s} f_t(\mathbf{x}_t) - \sum_{t=r}^{s} f_t(\mathbf{u})$$

$$\le \mathcal{O}\left(\sqrt{|I| \log T} + \log |I| \log T\right) = \mathcal{O}\left(\sqrt{(|I| + \log T \cdot \log^2 |I|) \log T}\right) = \mathcal{O}(\sqrt{|I| \log T}).$$

The last step holds by considering the following cases.

- When the interval length is $|I| = \Theta(T^\alpha)$ with $\alpha \in (0, 1]$. Then,

$$\mathcal{O}\left(\sqrt{(|I| + \log T \cdot \log^2 |I|) \log T}\right)$$

$$= \mathcal{O}\left(\sqrt{(T^\alpha + \alpha^2 \log^3 T) \log T}\right)$$

$$= \mathcal{O}\left(\sqrt{T^\alpha \log T}\right) = \mathcal{O}(\sqrt{|I| \log T}).$$

- When the interval length is $|I| = \Theta(\log^\beta T)$, and note that $\beta \in [1, +\infty)$ as $|I| = \Omega(\log T)$ is the minimum order to ensure the adaptive regret to be non-trivial. Then,

$$\mathcal{O}\left(\sqrt{(|I| + \log T \cdot \log^2 |I|) \log T}\right)$$

$$= \mathcal{O}\left(\sqrt{\left(\log^{\beta} T + \beta^2 \log T \cdot (\log \log T)^2\right) \log T}\right)$$

$$= \mathcal{O}\left(\sqrt{(\log^{\beta} T + \beta^2 \log T) \log T}\right)$$

$$= \mathcal{O}\left(\sqrt{\log^{\beta} T \log T}\right) = \mathcal{O}\left(\sqrt{|I| \log T}\right),$$

where the second equality is true for $\beta > 1$ and otherwise we can treat $\mathcal{O}(\log \log T)$ as a constant following previous studies [Luo and Schapire, 2015; Gaillard et al., 2014].

Hence we finish the proof for the worst-case adaptive regret bound. Combining both small-loss bound and the worst-case safety guarantee, we complete the proof of Theorem 4.

**Discussions about the analysis of worst-case bound.** We point out that the same analysis for worst-case adaptive bound cannot be applied to SACS [Zhang et al., 2019] directly, because the new designed *time-varying thresholds* mechanism plays an important role. Indeed, SACS monitors the cumulative loss of the new-added base-learner and switches to the another one immediately after a new marker is registered, which causes intermittent performance monitoring for the remaining markers the base-learner may go through. Thus, we cannot employ a similar argument (like Eq. (28) in our analysis) to bound the cumulative loss by the summation of thresholds for adaptive regret of SACS, as the performance of base-learner on the whole active interval is absent.

Finally, we would like to emphasize the necessity of monitoring the final decisions' cumulative loss. To meet the one projection requirement, our base-algorithm is updated over the surrogate loss and the provided gradient information is about the final decision only. Thus the thresholds are set on the cumulative loss of final decisions to exploit the limited available information. $\qquad \square$

### C.4 Proof of Lemma 4

*Proof.* First we introduce some useful variables to help us prove the adaptivity of Adapt-ML-Prod under sleeping-expert setting. Similar to the proof technique proposed in [Daniely et al., 2015], for any interval $[i, j] \in \widetilde{\mathcal{C}}$ in the geometric covers defined in (10), on which we suppose $m$-th base-learner is active, we define the following pseudo-weight for the $m$-th base-learner,

$$\widetilde{w}_{\tau,m} = \begin{cases} 0 & \tau < i, \\ 1 & \tau = i, \\ \left(\widetilde{w}_{\tau-1,m}(1 + \eta_{\tau-1}(\widehat{\ell}_{\tau-1} - \ell_{\tau-1,m}))\right)^{\frac{\eta_{\tau,m}}{\eta_{\tau-1,m}}} & i < \tau \le j+1, \\ \widetilde{w}_{j+1,m} & \tau > j+1. \end{cases}$$

In addition, we use $\widetilde{W}_t = \sum_{k \in [T]} \widetilde{w}_{t,k}$ to denote the summation of pseudo-weights for all possible base-learners up to time $t$. As for the problem-dependent geometric covers, in the worst case there are at most $T$ base-learners generated, we use $[T]$ to denote the indexes for all the base-learners. Notice that the pseudo-weight $\widetilde{w}_t$ is defined as $0$ for asleep base-learners till time $t$, so we can include all possible ones safely in the definition even though they are not generated in practical implementations of the algorithm.

In the following, we use the classic potential argument [Gaillard et al., 2014] by showing both lower bound and upper bound of $\ln \widetilde{W}_{t+1}$ to establish relationships between certain concerned quantities.

**Lower bound of $\ln \widetilde{W}_{t+1}$.** We claim that for $t \in [i, j]$ it holds that

$$\ln \widetilde{w}_{t+1,m} \ge \eta_{t+1,m} \sum_{\tau=i}^{t} (r_{\tau,m} - \eta_{\tau,m} r_{\tau,m}^2). \tag{29}$$

We prove the above inequality by induction on $t$. When $t = i$, by definition,

$$\ln \widetilde{w}_{i+1,m} = \frac{\eta_{i+1,m}}{\eta_{i,m}} \ln(1 + \eta_m r_{i,m}) \ge \frac{\eta_{i+1,m}}{\eta_{i,m}} \left(\eta_m r_{i,m} - \eta_m^2 r_{i,m}^2\right) = \eta_{i+1,m}(r_{i,m} - \eta_m r_{i,m}^2),$$

where the inequality is because of $\ln(1+x) \geq x - x^2, \forall x \geq -1/2$.

Suppose the statement holds for $\ln \widetilde{w}_{t,m}$, then we proceed to check the situation for $t+1$ round as follows. Indeed,

$$
\begin{aligned}
\ln \widetilde{w}_{t+1,m} &= \frac{\eta_{t+1,m}}{\eta_{t,m}} \left( \ln \widetilde{w}_{t,m} + \ln\left(1 + \eta_{t,m} r_{t,m}\right) \right) \\
&\geq \frac{\eta_{t+1,m}}{\eta_{t,m}} \left( \ln \widetilde{w}_{t,m} + \eta_{t,m} r_{t,m} - \eta_{t,m}^2 r_{t,m}^2 \right) \\
&= \frac{\eta_{t+1,m}}{\eta_{t,m}} \ln \widetilde{w}_{t,m} + \eta_{t+1,m} \left( r_{t,m} - \eta_{t,m} r_{t,m}^2 \right) \\
&\geq \frac{\eta_{t+1,m}}{\eta_{t,m}} \left( \eta_{t,m} \sum_{\tau=i}^{t-1} (r_{\tau,m} - \eta_{\tau,m} r_{\tau,m}^2) \right) + \eta_{t+1,m} \left( r_{t,m} - \eta_{t,m} r_{t,m}^2 \right) \\
&= \eta_{t+1,m} \sum_{\tau=i}^{t} (r_{\tau,m} - \eta_{\tau,m} r_{\tau,m}^2). \tag{30}
\end{aligned}
$$

Then, as $\widetilde{w}_{t+1,m}$ is positive for any $m$-th base-learner, we have $\ln \widetilde{W}_{t+1} \geq \ln \widetilde{w}_{t+1,m}$. Combining (30) obtains the desired lower bound of $\ln \widetilde{W}_{t+1}$.

**Upper bound of $\ln \widetilde{W}_{t+1}$.** By the construction of the geometric covers as specified in Eq. (10), we know that there will be at most $2m$ base-learners initialized for the $m$-th base-learner active on interval $[i, j]$ till her end. This is because $m$-th base-learner is initialized when $m$-th marker is recorded, and she will expire before the moment when $2m$-th marker is recorded, as demonstrated by the construction of cover defined in Eq. (10). Owing to this property, we have $\widetilde{W}_{t+1} = \sum_{k \in [2m]} \widetilde{w}_{t+1,k}$ as others' pseudo-weight equals to $0$ by definition. So we can upper bound $\widetilde{W}_{t+1}$ as,

$$
\begin{aligned}
\widetilde{W}_{t+1} = \sum_{k \in [2m]} \widetilde{w}_{t+1,k} &= \sum_{k \in [2m]: i_k = t+1} \widetilde{w}_{t+1,k} + \sum_{k \in [2m]: i_k \leq t} \widetilde{w}_{t+1,k} \\
&= \mathbb{1}\{\text{new alg. at } t+1\} + \sum_{k \in [2m]: i_k \leq t} \widetilde{w}_{t+1,k}, \tag{31}
\end{aligned}
$$

where with a slight abuse of notations, we denote by $[i_k, j_k] \in \widehat{\mathcal{C}}$ the active time for $k$-th base-learner. For the second term in (31), we have

$$
\begin{aligned}
\sum_{k: i_k \leq t} \widetilde{w}_{t+1,k} &= \sum_{k \in [2m]: t \in [i_k, j_k]} \widetilde{w}_{t+1,k} + \sum_{k \in [2m]: t > j_k} \widetilde{w}_{t+1,k} \\
&= \sum_{k \in [2m]: t \in [i_k, j_k]} \widetilde{w}_{t+1,k} + \sum_{k \in [2m]: t > j_k} \widetilde{w}_{t,k} \\
&\leq \sum_{k \in [2m]: t \in [i_k, j_k]} \widetilde{w}_{t,k}(1 + \eta_{t,k} r_{t,k}) + \frac{1}{e}\left(\frac{\eta_{t,k}}{\eta_{t+1,k}} - 1\right) + \sum_{k \in [2m]: t > j_k} \widetilde{w}_{t,k} \\
&= \widetilde{W}_t + \underbrace{\sum_{k \in [2m]: t \in [i_k, j_k]} \eta_{t,k} \widetilde{w}_{t,k} r_{t,k}}_{=0} + \sum_{k \in [2m]: t \in [i_k, j_k]} \frac{1}{e}\left(\frac{\eta_{t,k}}{\eta_{t+1,k}} - 1\right), \tag{32}
\end{aligned}
$$

where the first equality holds by the definition of $\widetilde{w}_{t+1,k}$, the second inequality is by the updating rule of $\widetilde{w}_{t+1,k}$ and Lemma (18), and the second term in the last equality equals to $0$ due to the weight update rule in (13) and the fact of $\widetilde{w}_{t,k} = w_{t,k}$ for any $t \in [i_k, j_k]$.

Combining (31), (32) and by induction, we obtain the following upper bound:

$$
\widetilde{W}_{t+1} \leq 1 + 2m + \frac{1}{e} \sum_{k \in [2m]} \sum_{\tau=i_k}^{t \wedge j_k} \left(\frac{\eta_{\tau,k}}{\eta_{\tau+1,k}} - 1\right), \tag{33}
$$

where we denote $\alpha \wedge \beta = \min\{\alpha, \beta\}$.

We now turn to analyze the third term in (32). Indeed, Gaillard et al. [2014] have analyzed it under the static regret measure. For the sake of completeness, we present the proof with our notations. For any $k \in [2m]$, for any $\tau \in [i_k, t \wedge j_k]$, the relationship between $\eta_{\tau,k}$ and $\eta_{\tau+1,k}$ can be considered as three cases,

- $\eta_{\tau,k} = \eta_{\tau+1,k} = 1/2$,

- $\eta_{\tau+1,k} = \sqrt{\gamma_k / (1 + \sum_{u=i_k}^{\tau} r_{u,k}^2)} < \eta_{\tau,k} = \frac{1}{2}$,

- $\eta_{\tau+1,k} \leq \eta_{\tau,k} < 1/2$.

In all cases, the ratio $\eta_{\tau,k} / \eta_{\tau+1,k} - 1$ is at most

$$
\sum_{\tau=i_k}^{t \wedge j_k} \left( \frac{\eta_{\tau,k}}{\eta_{\tau+1,k}} - 1 \right) \leq \sum_{\tau=i_k}^{t \wedge j_k} \left( \sqrt{\frac{1 + \sum_{u=i_k}^{\tau} r_{u,k}^2}{1 + \sum_{u=i_k}^{\tau-1} r_{u,k}^2}} - 1 \right)
$$

$$
= \sum_{\tau=i_k}^{t \wedge j_k} \left( \sqrt{\frac{r_{\tau,k}^2}{1 + \sum_{u=i_k}^{\tau-1} r_{u,k}^2} + 1} - 1 \right)
$$

$$
\leq \frac{1}{2} \sum_{\tau=i_k}^{t \wedge j_k} \frac{r_{\tau,k}^2}{1 + \sum_{u=i_k}^{\tau-1} r_{u,k}^2}
$$

$$
\leq \frac{1}{2} \left( 1 + \ln \left( 1 + \sum_{u=i_k}^{t \wedge j_k} r_{u,k}^2 \right) \right) - \ln(1)
$$

$$
\leq \frac{1}{2} \left( 1 + \ln(1 + t) \right), \tag{34}
$$

where the second inequality uses $\sqrt{1 + x} \leq 1 + x/2$ and the third inequality follows from Lemma 14 with the choice of $f(x) = 1/x$.

Substituting (34) into (33), we get

$$
\widetilde{W}_{t+1} \leq 1 + 2m + \frac{m}{e} \left( 1 + \ln(1 + t) \right) \leq (1 + 2m) \left( 1 + \frac{1}{2e} \left( 1 + \ln(1 + t) \right) \right). \tag{35}
$$

Further taking the logarithm over the above inequality gives the upper bound of $\ln \widetilde{W}_{t+1}$.

**Upper bound of meta-regret.** Now, we can lower bound and upper bound $\ln \widetilde{W}_{t+1}$ by (30) and (35), with arrangement, which yields the upper bound of scaled meta-regret. Concretely,

$$
\sum_{\tau=i}^{t} r_{\tau,m} \leq \sum_{\tau=i}^{t} \eta_{\tau,m} r_{\tau,m}^2 + \frac{\ln(1 + 2m) + \mu(t)}{\eta_{t+1,m}}
$$

$$
\leq 2\sqrt{\gamma_i} \sqrt{1 + \sum_{\tau=i}^{t} r_{\tau,i}^2} + \frac{\ln(1 + 2m) + \mu(t)}{\eta_{t+1,m}} \tag{36}
$$

$$
\leq \frac{\ln(1 + 2m) + \mu(t) + 2\gamma_m}{\sqrt{\gamma_m}} \sqrt{1 + \sum_{\tau=i}^{t} r_{\tau,m}^2 + 2\ln(1 + 2m) + 4\gamma_m + 2\mu(t)}
$$

$$
= \left( 3\sqrt{\ln(1 + 2m)} + \frac{\mu(t)}{\sqrt{\ln(1 + 2m)}} \right) \sqrt{1 + \sum_{\tau=i}^{t} r_{\tau,m}^2 + 6\ln(1 + 2m) + 2\mu(t)}, \tag{37}
$$

where we denote $\mu(t) = \ln(1 + (1 + \ln(1 + t))/(2e))$. The second inequality is by Lemma 14 and choose $f(x) = 1/\sqrt{x}$. The last equality is by the choice of $\gamma_m = \ln(1 + 2m)$. As for the third inequality, there are two cases to be considered:

- when $\sqrt{1 + \sum_{\tau=i}^{t} r_{\tau,m}^2} > 2\sqrt{\gamma_m}$, we have that (36) is at most

$$2\sqrt{\gamma_m}\sqrt{1 + \sum_{\tau=i}^{t} r_{\tau,m}^2} + \frac{\ln(1+2m) + \mu(t)}{\sqrt{\gamma_m}}\sqrt{1 + \sum_{\tau=i}^{t} r_{\tau,m}^2}.$$

- when $\sqrt{1 + \sum_{\tau=i}^{t} r_{\tau,m}^2} \le 2\sqrt{\gamma_m}$, we have that $\eta_{t+1,m} = 1/2$ and (36) is at most

$$2\ln(1+2m) + 4\gamma_m + 2\mu(t).$$

Taking both cases into account implies the third inequality.

Finally, we end the proof by evaluating the meta-regret in terms of the surrogate loss.

$$\sum_{\tau=i}^{t} \langle \nabla g_t(\mathbf{y}_t), \mathbf{y}_t - \mathbf{y}_{t,m} \rangle$$

$$= 2GD \cdot \sum_{\tau=i}^{t} r_{\tau,m}$$

$$\le 2GD\left(3\sqrt{\ln(1+2m)} + \frac{\mu(t)}{\sqrt{\ln(1+2m)}}\right)\sqrt{1 + \sum_{\tau=i}^{t} r_{\tau,m}^2} + 12GD\ln(1+2m) + 4GD\mu(t)$$

$$\le \left(3\sqrt{\ln(1+2m)} + \frac{\mu(t)}{\sqrt{\ln(1+2m)}}\right)\sqrt{\sum_{\tau=i}^{t} \langle \nabla g_\tau(\mathbf{y}_\tau), \mathbf{y}_\tau - \mathbf{y}_{\tau,m} \rangle^2} + 18GD\ln(1+2m) + 6GD\mu(t)$$

$$\le \left(3\sqrt{\ln(1+2m)} + \frac{\mu(t)}{\sqrt{\ln(1+2m)}}\right)\sqrt{\sum_{\tau=i}^{t} 4D^2\|\nabla g_\tau(\mathbf{y}_\tau)\|_2^2} + 18GD\ln(1+2m) + 6GD\mu(t)$$

$$\le 2D\left(3\sqrt{\ln(1+2m)} + \frac{\mu(t)}{\sqrt{\ln(1+2m)}}\right)\sqrt{\sum_{\tau=i}^{t} \|\nabla f_\tau(\mathbf{x}_\tau)\|_2^2} + 18GD\ln(1+2m) + 6GD\mu(t)$$

$$\tag{38}$$

$$\le 4D\left(3\sqrt{\ln(1+2m)} + \frac{\mu(t)}{\sqrt{\ln(1+2m)}}\right)\sqrt{L\sum_{\tau=i}^{t} f_t(\mathbf{x}_t)} + 18GD\ln(1+2m) + 6GD\mu(t),$$

$$\tag{39}$$

where the second inequality is true because $1 \le \sqrt{\ln(1+2m)} \le \ln(1+2m)$ holds for any $m \ge 1$, the third inequality is by Cauchy-Schwarz inequality, the forth inequality is by Theorem 1 and the last inequality is due to the self-bounded property of smooth functions (see Lemma 12). □

### C.5  Proof of Lemma 5

*Proof.* Similar to the proof of dynamic regret (see Theorem 3), we start the proof by decomposing the interval regret into meta-regret and base-regret in terms of the surrogate loss by Theorem 1,

$$\sum_{\tau=i}^{t} f_\tau(\mathbf{x}_\tau) - \sum_{t=i}^{t} f_\tau(\mathbf{u}) \le \sum_{\tau=i}^{t} g_\tau(\mathbf{x}_\tau) - \sum_{t=i}^{t} g_\tau(\mathbf{u}) \le \sum_{\tau=i}^{t} \langle \nabla g_\tau(\mathbf{y}_\tau), \mathbf{y}_\tau - \mathbf{u} \rangle$$

$$= \underbrace{\sum_{\tau=i}^{t} \langle \nabla g_\tau(\mathbf{y}_\tau), \mathbf{y}_\tau - \mathbf{y}_{\tau,m} \rangle}_{\texttt{meta-regret}} + \underbrace{\sum_{\tau=i}^{t} \langle \nabla g_\tau(\mathbf{y}_\tau), \mathbf{y}_{\tau,m} - \mathbf{u} \rangle}_{\texttt{base-regret}}, \tag{40}$$

where our analysis will be performed by tracking the $m$-th base-learner, whose corresponding active interval is exactly the analyzed one. Our analysis is satisfied to any interval since there is always a base-learner active on it by the algorithm design.

**Upper bound of base-regret.** Since the base-algorithm (SOGD) guarantees anytime regret, direct application of Lemma 8 with the assumption of surrogate domain $\mathcal{Y}$ can upper bound the base-regret,

$$\sum_{\tau=i}^{t} \langle \nabla g_\tau(\mathbf{y}_\tau), \mathbf{y}_{\tau,m} - \mathbf{u} \rangle \leq 4D \sqrt{\delta + \sum_{\tau=i}^{t} \|\nabla g_\tau(\mathbf{y}_\tau)\|_2^2} \leq 8D \sqrt{L \sum_{\tau=i}^{t} f_\tau(\mathbf{x}_\tau) + 4D\sqrt{\delta}}, \quad (41)$$

where we skip some steps for transforming $\|\nabla g_\tau(\mathbf{y}_\tau)\|_2^2$ into $4L f_\tau(\mathbf{x}_\tau)$. The similar arguments can be found in the proof of Theorem 3.

**Upper bound of meta-regret.** By Lemma 4 (see Eq. (39) for a detailed form), we can upper bound the meta-regret as

$$\sum_{\tau=i}^{t} \langle \nabla g_\tau(\mathbf{y}_\tau), \mathbf{y}_\tau - \mathbf{y}_{\tau,m} \rangle$$

$$\leq 4D \left( 3\sqrt{\ln(1+2m)} + \frac{\mu(t)}{\sqrt{\ln(1+2m)}} \right) \sqrt{L \sum_{\tau=i}^{t} f_\tau(\mathbf{x}_\tau) + 18GD \ln(1+2m) + 6GD\mu(t)},$$

$$(42)$$

where we denote $\mu(t) = \ln(1 + (1 + \ln(1+t))/(2e))$, which is of order $\mathcal{O}(\log\log t)$ and can be treated as a constant.

**Upper bound of interval regret.** Substituting (41), (42) into (40) obtains the interval regret

$$\sum_{\tau=i}^{t} f_\tau(\mathbf{x}_\tau) - \sum_{\tau=i}^{t} f_\tau(\mathbf{u}) \leq \mathcal{O} \left( \sqrt{\log m \cdot \sum_{\tau=i}^{t} f_\tau(\mathbf{x}_\tau)} + \log m \right), \quad (43)$$

which is related to the returned decisions of our algorithm.

Further, by applying Lemma 17 to the preceding inequality, we can substitute $\sum_{\tau=i}^{t} f_\tau(\mathbf{x}_\tau)$ into decision-independent factor $\sum_{\tau=i}^{t} f_\tau(\mathbf{u})$,

$$\sum_{\tau=i}^{t} f_\tau(\mathbf{x}_\tau) - \sum_{\tau=i}^{t} f_\tau(\mathbf{u})$$

$$\leq 4D\sqrt{L} \left( \sqrt{\ln(1+2m)} + \frac{\mu(t)}{\sqrt{\ln(1+2m)}} + 2 \right) \sqrt{\sum_{\tau=i}^{t} f_\tau(\mathbf{u}) + 18GD\ln(1+2m) + 6GD\mu(t) + 4D\sqrt{\delta}}$$

$$\quad + 18GD\ln(1+2m) + 6GD\mu(t) + 4D\sqrt{\delta} + 16D^2 L \left( \sqrt{\ln(1+2m)} + \frac{\mu(t)}{\sqrt{\ln(1+2m)}} + 2 \right)^2$$

$$\leq 4D\sqrt{L} \left( \sqrt{\ln(1+2m)} + \frac{\mu(t)}{\sqrt{\ln(1+2m)}} + 2 \right) \sqrt{\sum_{\tau=i}^{t} f_\tau(\mathbf{u})}$$

$$\quad + 27GD\ln(1+2m) + 9GD\mu(t) + 6D\sqrt{\delta} + 24D^2 L \left( \sqrt{\ln(1+2m)} + \frac{\mu(t)}{\sqrt{\ln(1+2m)}} + 2 \right)^2$$

$$\leq 4D\sqrt{L} \left( \sqrt{\ln(1+2m)} + \frac{\mu(t)}{\sqrt{\ln(1+2m)}} + 2 \right) \sqrt{\sum_{\tau=i}^{t} f_\tau(\mathbf{u})}$$

$$\quad + (27GD + 72D^2 L)\ln(1+2m) + 72D^2 L\mu^2(t) + 9GD\mu(t) + 6D\sqrt{\delta} + 288D^2 L. \quad (44)$$

The second inequality makes use of $\sqrt{a+b} \leq \sqrt{a} + \sqrt{b}$ and $\sqrt{ab} \leq (a^2 + b^2)/2$. The last inequality holds by $(a+b+c)^2 \leq 3(a^2+b^2+c^2)$.

Finally, with a slight abuse of notations, we show that actually the interval regret can be related to the best offline cumulative loss, i.e., the quantity $F_{[i,t]} = \min_{\mathbf{u} \in \mathcal{X}} \sum_{\tau=i}^{t} f_\tau(\mathbf{u})$. Notice that, the above

arguments hold for any $\mathbf{u} \in \mathcal{X}$, so we can simply choose $\mathbf{u}^* \in \arg\min_{\mathbf{u}' \in \mathcal{X}} \sum_{\tau=i}^{t} f_\tau(\mathbf{u}')$ as the comparator in the above arguments, and for any $\mathring{\mathbf{u}} \in \mathcal{X}$,

$$\sum_{\tau=i}^{t} f_\tau(\mathbf{x}_\tau) - \sum_{\tau=i}^{t} f_\tau(\mathring{\mathbf{u}}) \leq \sum_{\tau=i}^{t} f_\tau(\mathbf{x}_\tau) - \min_{\mathbf{u} \in \mathcal{X}} \sum_{\tau=i}^{t} f_\tau(\mathbf{u}) \leq \mathcal{O}\left(\sqrt{\log m \cdot F_{[i,t]}}\right).$$

This ends the proof. $\qquad\square$

## C.6    Proof of Lemma 6

*Proof.* We claim that the cumulative loss of our algorithm within every interval in the geometric covers $\widetilde{\mathcal{C}}$ as defined in Eq. (10) can be upper bounded by the cumulative loss of any comparator and the threshold set for the interval.

To see this, we denote by $[i_k, j_k] \in \widetilde{\mathcal{C}}$ the active interval for the $k$-th base-learner. By Lemma 5, for any $t' \in [i_k, j_k]$ we have

$$\sum_{\tau=i_k}^{t'} f_\tau(\mathbf{x}_\tau) \leq 2 \sum_{\tau=i_k}^{t'} f_\tau(\mathbf{u}) + \frac{1}{2}\mathcal{G}(k), \tag{45}$$

where $\mathcal{G}(k)$ is the threshold generating function defined by

$$\mathcal{G}(k) = (54GD + 168D^2 L)\ln(1+2k) + 168D^2 L\mu^2(T) + 18GD\mu(T) + 6D\sqrt{\delta} + 672D^2 L, \tag{46}$$

with $\mu(T) = \ln(1 + \frac{1+\ln(1+T)}{2e})$. This is true by applying $ab \leq a^2/4 + b^2$ and $(a+b+c)^2 \leq 3(a^2+b^2+c^2)$ to split $\sum f_t(\mathbf{u})$ outside the root in (44).

We denote $s_k$ the $k$-th marker made by the algorithm and it is known that $i_k = s_k$ (but $s_{k+1} - 1 \leq j_k$ because the base-learner may survive several markers) by the cover mechanism. According to our algorithm design, we set the threshold for interval $[s_k, s_{k+1} - 1]$ as

$$C_k = \mathcal{G}(k).$$

By the construction rule of the markers, we have,

$$\sum_{\tau=s_k}^{s_{k+1}-1} f_\tau(\mathbf{x}_\tau) \geq C_k.$$

For the $k$-th base-learner, her ending time $j_k$ should be equal or larger than $s_{k+1} - 1$, so we can use (45) to evaluate the lower bound of the cumulative loss of the comparator by setting $t' = s_{k+1} - 1$. Indeed, we have

$$\sum_{\tau=s_k}^{s_{k+1}-1} f_\tau(\mathbf{u}) \geq \frac{1}{2}\left(\sum_{\tau=s_k}^{s_{k+1}-1} f_\tau(\mathbf{x}_\tau) - \frac{1}{2}\mathcal{G}(k)\right) \geq \frac{1}{2}\left(C_k - \frac{1}{2}\mathcal{G}(k)\right) = \frac{1}{2}\left(C_k - \frac{1}{2}C_k\right) = \frac{1}{4}C_k. \tag{47}$$

It is worth emphasizing that, after making each marker, the cover will initialize a new base-learner and hence the evaluation as shown above can be made between every two consecutive markers thanks to the streaming initialized new learners.

Next, we proceed to upper bound $m$ given in the lemma. As the $m$-th base-learner is active on interval $[i, j]$, we have the following result on the interval from marker $s_1$ to $s_m$,

$$\sum_{\tau=s_1}^{s_m-1} f_\tau(\mathbf{u}) \geq \frac{1}{4}\sum_{a=1}^{m-1} C_a \geq \frac{C_1}{4}(m-1).$$

The first inequality is because of (47). The second inequality holds since $C_a$ is increasing with respect to its index, see the threshold generating function in Eq. (46).

Therefore, rearranging the above inequality shows that the quantity $m$ satisfies the following inequality for any comparator $\mathbf{u} \in \mathcal{X}$,

$$m \leq 1 + \frac{4}{C_1}\sum_{\tau=s_1}^{s_m-1} f_\tau(\mathbf{u}) \leq 1 + \frac{4}{C_1}\sum_{\tau=1}^{t} f_\tau(\mathbf{u}),$$

where the last inequality makes use of the nonnegative assumption on loss function. In particular, we choose the comparator to be the best offline decision optimizing the cumulative loss within this interval and thus achieve

$$m \leq 1 + \frac{4}{C_1} \min_{\mathbf{u} \in \mathcal{X}} \sum_{\tau=1}^{t} f_\tau(\mathbf{u}) = \mathcal{O}(F_{[1,t]}), \tag{48}$$

where we denote $F_{[1,t]} = \min_{\mathbf{u} \in \mathcal{X}} \sum_{\tau=1}^{t} f_\tau(\mathbf{u})$.

Now combining Lemma 5 and (48), once given time $t$, we can upper bound the number of base-learners by the cumulative loss of comparators,

$$\sum_{\tau=i}^{t} f_\tau(\mathbf{x}_\tau) - \sum_{\tau=i}^{t} f_\tau(\mathbf{u}) \leq \mathcal{O}\left(\sqrt{\log(m) F_{[i,t]}}\right) = \mathcal{O}\left(\sqrt{\ln(F_{[1,t]}) \cdot F_{[i,t]}}\right),$$

which ends the proof of Lemma 6. $\qquad\square$

# D Useful Lemmas

This section collects some lemmas useful for the proofs.

## D.1 OGD and Dynamic Regret

This part provides the dynamic regret of online gradient descent (OGD) [Zinkevich, 2003] and scale-free online gradient descent (SOGD) [Orabona and Pál, 2018] from the view of online mirror descent (OMD), which is a common and powerful online learning framework. Following the analysis in [Zhao et al., 2021b], we can directly obtain dynamic regret of OGD and SOGD [Orabona and Pál, 2018] in a unified view owing to the versatility of OMD.

Online mirror descent algorithm updates according to the following rule:

$$\mathbf{x}_{t+1} = \underset{\mathbf{x} \in \mathcal{X}}{\arg\min} \; \eta_t \langle \nabla f_t(\mathbf{x}_t), \mathbf{x} \rangle + \mathcal{D}_\psi(\mathbf{x}, \mathbf{x}_t), \tag{49}$$

where $\eta_t > 0$ is the time-varying step size, $h_t(\cdot) : \mathbf{x} \mapsto \mathbb{R}$ is the convex loss function, and $\mathcal{D}_\psi(\cdot, \cdot)$ is the Bregman divergence induced by the regularizer function $\psi(\cdot)$ defined as $\mathcal{D}_\psi(\mathbf{x}, \mathbf{y}) = \psi(\mathbf{x}) - \psi(\mathbf{y}) - \langle \nabla \psi(\mathbf{y}), \mathbf{x} - \mathbf{y} \rangle$. OMD following dynamic regret guarantee [Zhao et al., 2021b].

**Theorem 5** (Theorem 1 of Zhao et al. [2021b]). *Suppose that the regularizer $\psi : \mathcal{X} \mapsto \mathbb{R}$ is 1-strongly convex with respect to the norm $\|\cdot\|$. The dynamic regret of Optimistic Mirror Descent (OMD) whose update rule specified in* (49) *is bounded as follows:*

$$\sum_{t=1}^{T} f_t(\mathbf{x}_t) - \sum_{t=1}^{T} f_t(\mathbf{u}_t) \leq \sum_{t=1}^{T} \eta_t \|\nabla f_t(\mathbf{x}_t)\|_*^2 + \sum_{t=1}^{T} \frac{1}{\eta_t} \left( \mathcal{D}_\psi(\mathbf{u}_t, \mathbf{x}_t) - \mathcal{D}_\psi(\mathbf{u}_t, \mathbf{x}_{t+1}) \right)$$
$$- \sum_{t=1}^{T} \frac{1}{\eta_t} \mathcal{D}_\psi(\mathbf{x}_{t+1}, \mathbf{x}_t),$$

*which holds for any comparator sequence $\mathbf{u}_1, \ldots, \mathbf{u}_T \in \mathcal{X}$.*

Choosing $\psi(\mathbf{x}) = \frac{1}{2} \|\mathbf{x}\|_2^2$ will lead to the update form of online gradient descent used as base learners in our algorithm:

$$\mathbf{x}_{t+1} = \underset{\mathbf{x} \in \mathcal{X}}{\arg\min} \; \eta_t \langle \nabla f_t(\mathbf{x}_t), \mathbf{x} \rangle + \frac{1}{2} \|\mathbf{x} - \mathbf{x}_t\|_2^2, \tag{50}$$

where the Bregman divergence becomes $\mathcal{D}_\psi(\mathbf{x}, \mathbf{x}_t) = \frac{1}{2} \|\mathbf{x} - \mathbf{x}_t\|_2^2$ w.r.t. the choice of regularizer.

We proceed to show the dynamic regret of online gradient descent (OGD),

**Lemma 7.** *Under Assumption 2, by choosing static step size $\eta_t = \eta > 0$, Online Gradient Descent defined in equation* (50) *satisfies:*

$$\sum_{t=1}^{T} f_t(\mathbf{x}_t) - \sum_{t=1}^{T} f_t(\mathbf{u}_t) \leq \frac{7D^2}{4\eta} + \frac{D}{\eta} \sum_{t=2}^{T} \|\mathbf{u}_{t-1} - \mathbf{u}_2\|_2 + \eta \sum_{t=1}^{T} \|\nabla f_t(\mathbf{x}_t)\|_2^2$$

*for any comparator sequence $\mathbf{u}_1, \ldots, \mathbf{u}_T \in \mathcal{X}$.*

*Proof.* Applying Theorem 5 with the choices of $\psi(\mathbf{x}) = \frac{1}{2}\|\mathbf{x}\|_2^2$ and fixed step size $\eta_t = \eta > 0$ gives:

$$\sum_{t=1}^{T} f_t(\mathbf{x}_t) - \sum_{t=1}^{T} f_t(\mathbf{u}_t)$$

$$\leq \frac{1}{2\eta} \sum_{t=1}^{T} \left( \|\mathbf{u}_t - \mathbf{x}_t\|_2^2 - \|\mathbf{u}_t - \mathbf{x}_{t+1}\|_2^2 \right) + \eta \sum_{t=1}^{T} \|\nabla f_t(\mathbf{x}_t)\|_2^2 - \frac{1}{2\eta} \sum_{t=1}^{T} \|\mathbf{x}_t - \mathbf{x}_{t+1}\|_2^2$$

$$\leq \frac{1}{2\eta} \sum_{t=1}^{T} \left( \|\mathbf{x}_t\|_2^2 - \|\mathbf{x}_{t+1}\|_2^2 \right) + \frac{1}{\eta} \sum_{t=1}^{T} (\mathbf{x}_{t+1} - \mathbf{x}_t)^\top \mathbf{u}_t + \eta \sum_{t=1}^{T} \|\nabla f_t(\mathbf{x}_t)\|_2^2$$

$$\leq \frac{1}{2\eta} \|\mathbf{x}_1\|_2^2 + \frac{1}{\eta} \left( \mathbf{x}_{T+1}^\top \mathbf{u}_T - \mathbf{x}_1^\top \mathbf{u}_1 \right) + \frac{1}{\eta} \sum_{t=2}^{T} (\mathbf{u}_{t-1} - \mathbf{u}_t)^\top \mathbf{x}_t + \eta \sum_{t=1}^{T} \|\nabla f_t(\mathbf{x}_t)\|_2^2$$

$$\leq \frac{7D^2}{4\eta} + \frac{D}{\eta} \sum_{t=2}^{T} \|\mathbf{u}_{t-1} - \mathbf{u}_2\|_2 + \eta \sum_{t=1}^{T} \|\nabla f_t(\mathbf{x}_t)\|_2^2,$$

where the last inequality is due to:

$$\|\mathbf{x}_1\|_2^2 = \|\mathbf{x}_1 - \mathbf{0}\|_2^2 \leq D^2,$$
$$\mathbf{x}_{T+1}^\top \mathbf{u}_T \leq \|\mathbf{x}_{T+1}\|_2 \cdot \|\mathbf{u}_T\|_2 = D^2,$$
$$-\mathbf{x}_1^\top \mathbf{u}_1 \leq \frac{1}{4}\|\mathbf{x}_1 - \mathbf{u}_1\|_2^2 \leq \frac{1}{4}D^2,$$
$$(\mathbf{u}_{t-1} - \mathbf{u}_t)^\top \mathbf{x}_t \leq \|\mathbf{u}_{t-1} - \mathbf{u}_t\|_2 \cdot \|\mathbf{x}_t\|_2 \leq D\|\mathbf{u}_{t-1} - \mathbf{u}_t\|_2.$$

$\square$

## D.2  Self-Confident Tuning

Orabona and Pál [2018] have analyzed the regret bound of SOGD. For completeness, we here provide the regret analysis under the OMD framework. Indeed, SOGD can be treated as OMD with a self-confident learning rate. Thus, we have the following lemma.

**Lemma 8.** *Under assumption 1 and 2, the OMD algorithm defined in equation (49) with the choices of regularizer $\psi(\mathbf{x}) = \frac{1}{2}\|\mathbf{x}\|_2^2$ and the time-varying learning rate defined as*

$$\eta_t = \frac{D/2}{\sqrt{\delta + \sum_{\tau=1}^{t-1} \|\nabla f_t(\mathbf{x}_t)\|_2^2}}$$

*for some $\delta > 0$, has the following guarantee:*

$$\sum_{t=1}^{T} f_t(\mathbf{x}_t) - \sum_{t=1}^{T} f_t(\mathbf{u}) \leq 2D \cdot \sqrt{\delta + \sum_{t=1}^{T} \|\nabla f_t(\mathbf{x}_t)\|_2^2},$$

*where $\mathbf{u} \in \mathcal{X}$ can be any comparator.*

*Proof.* We start the proof with the application of Theorem 5. For any fixed comparator $\mathbf{u} \in \mathcal{X}$,

$$\sum_{t=1}^{T} f_t(\mathbf{x}_t) - \sum_{t=1}^{T} f_t(\mathbf{u}_t)$$

$$\leq \sum_{t=1}^{T} \frac{1}{2\eta_t} \left( \|\mathbf{u} - \mathbf{x}_t\|_2^2 - \|\mathbf{u} - \mathbf{x}_{t+1}\|_2^2 \right) + \sum_{t=1}^{T} \eta_t \|\nabla f_t(\mathbf{x}_t)\|_2^2 - \sum_{t=1}^{T} \frac{1}{2\eta_t} \|\mathbf{x}_t - \mathbf{x}_{t+1}\|_2^2$$

$$\leq \frac{1}{2\eta_1} \|\mathbf{u} - \mathbf{x}_1\|_2^2 + \sum_{t=2}^{T} \left( \frac{1}{\eta_t} - \frac{1}{\eta_{t-1}} \right) \frac{\|\mathbf{u} - \mathbf{x}_t\|_2^2}{2} + \sum_{t=1}^{T} \eta_t \|\nabla f_t(\mathbf{x}_t)\|_2^2$$

$$\leq \frac{D^2}{2\eta_1} + \frac{D^2}{2}\sum_{t=2}^{T}\left(\frac{1}{\eta_t} - \frac{1}{\eta_{t-1}}\right) + \sum_{t=1}^{T}\eta_t\|\nabla f_t(\mathbf{x}_t)\|_2^2$$

$$= \frac{D^2}{2\eta_T} + \sum_{t=1}^{T}\eta_t\|\nabla f_t(\mathbf{x}_t)\|_2^2. \tag{51}$$

Applying Lemma 13 to the second term of (51) and by the definition of $\eta_T$, the regret bound becomes

$$\sum_{t=1}^{T}f_t(\mathbf{x}_t) - \sum_{t=1}^{T}f_t(\mathbf{u}_t) \leq D \cdot \sqrt{\delta + \sum_{t=1}^{T}\|\nabla f_t(\mathbf{x}_t)\|_2^2} + D\left(\sqrt{\delta + \sum_{t=1}^{T}\|\nabla f_t(\mathbf{x}_t)\|_2^2} - \sqrt{\delta}\right)$$

$$\leq 2D \cdot \sqrt{\delta + \sum_{t=1}^{T}\|\nabla f_t(\mathbf{x}_t)\|_2^2},$$

which completes the proof. $\qquad\square$

To bound the meta-regret of our dynamic methods, we introduce the FTRL lemma [Orabona, 2019, Corollary 7.8] under the time-varying learning rate.

**Lemma 9** (FTRL Lemma). *Suppose that the regularizer function $\psi : \mathcal{X} \mapsto \mathbb{R}$ is $\alpha$-strongly convex with respect to the norm $\|\cdot\|$. Let $f_t$ be a sequence of convex loss functions and $\psi_t(\mathbf{x}) = \frac{1}{\eta_t}(\psi(\mathbf{x}) - \min_{\mathbf{x}'\in\mathcal{X}}\psi(\mathbf{x}'))$, where $\eta_{t+1} \leq \eta_t$, $t = 1,\ldots,T$. Then the decision sequence $\mathbf{x}_t$ generated by*

$$\mathbf{x}_t = \underset{\mathbf{x}\in\mathcal{X}}{\arg\min}\left(\psi_t(\mathbf{x}) + \sum_{\tau=1}^{t-1}f_t(\mathbf{x})\right),$$

*satisfies the following regret upper bound for any $\mathbf{u} \in \mathcal{X}$,*

$$\sum_{t=1}^{T}f_t(\mathbf{x}_t) - f_t(\mathbf{u}) \leq \frac{\psi(\mathbf{u}) - \min_{\mathbf{x}\in\mathcal{X}}\psi(\mathbf{x})}{\eta_{T+1}} + \frac{1}{2\alpha}\sum_{t=1}^{T}\eta_t\|\nabla f_t(\mathbf{x}_t)\|_*^2.$$

Based on the preceding lemma, we can derive the regret upper bound for Hedge algorithm with a self-confident learning rate.

**Lemma 10.** *Consider the prediction with expert advice setting with $N$ experts and the linear loss $f_t(\mathbf{x}) = \langle \boldsymbol{\ell}_t, \mathbf{x}\rangle$, where $\boldsymbol{\ell}_t \in \mathbb{R}^d$. Then the self-confident tuning Hedge, whose initial decision is $\boldsymbol{p}_1 = 1/N \cdot \mathbf{1}$ and update rules are*

$$p_{t+1,i} \propto \exp\left(\varepsilon_{t+1}\sum_{\tau=1}^{t}\ell_{\tau,i}\right) \text{ with } \varepsilon_{t+1} = \sqrt{\frac{\ln N}{1 + \sum_{\tau=1}^{t}\|\boldsymbol{\ell}_\tau\|_\infty^2}}$$

*ensures the following regret guarantee: for any $i \in [N]$*

$$\sum_{t=1}^{T}\langle \boldsymbol{p}_t, \boldsymbol{\ell}_t\rangle - \sum_{t=1}^{T}\ell_{t,i} \leq 3\sqrt{\ln N \cdot \left(1 + \sum_{t=1}^{T}\|\boldsymbol{\ell}_t\|_\infty^2\right)} + \frac{\sqrt{\ln N}}{2}\cdot\max_{t\in[T]}\|\boldsymbol{\ell}_t\|_\infty^2.$$

*Proof.* It is easy to verify that, the mentioned self-confident tuning Hedge can be treated as a special case of the time-varying FTRL algorithm by choosing $\psi(\boldsymbol{p}) = \sum_{s=1}^{N}p_s\ln p_s$, which is 1-strongly convex with respect to $\|\cdot\|_1$, and $\psi_t(\boldsymbol{p}) = \frac{1}{\varepsilon_t}\psi(\boldsymbol{p})$. Thus, by Lemma 9, we have

$$\sum_{t=1}^{T}\langle \boldsymbol{p}_t, \boldsymbol{\ell}_t\rangle - \sum_{t=1}^{T}\ell_{t,i} \leq \frac{\ln N}{\varepsilon_{T+1}} + \frac{1}{2}\sum_{t=1}^{T}\varepsilon_t\|\boldsymbol{\ell}_t\|_\infty^2$$

$$\leq \frac{\ln N}{\varepsilon_{T+1}} + \frac{\sqrt{\ln N}}{2}\cdot\left(4\sqrt{1 + \sum_{t=1}^{T}\|\boldsymbol{\ell}_t\|_\infty^2} + \max_{t\in[T]}\|\boldsymbol{\ell}_t\|_\infty^2\right)$$

$$= 3\sqrt{\ln N \cdot \left(1 + \sum_{t=1}^{T}\|\boldsymbol{\ell}_t\|_\infty^2\right)} + \frac{\sqrt{\ln N}}{2} \cdot \max_{t \in [T]}\|\boldsymbol{\ell}_t\|_\infty^2,$$

where the first inequality chooses $\mathbf{u}$ as the one-hot vector with all entries being $0$ except the $i$-th one as $1$, and second inequality is by Lemma 15. $\qquad\square$

### D.3 Facts on Geometric Covers

**Lemma 11** (Lemma 11 of Zhang et al. [2019]). *Let $[s_p, s_q] \subseteq [T]$ be an arbitrary interval that starts from a marker $s_p$ and ends at another marker $s_q$. Then, we can find a sequence of consecutive intervals*

$$I_1 = [s_{i_1}, s_{i_2} - 1],\ I_2 = [s_{i_2}, s_{i_3} - 1],\ \ldots, I_v = [s_{i_v}, s_{i_{v+1}} - 1] \in \widetilde{\mathcal{C}}$$

*such that*

$$i_1 = p,\ i_v \le q < i_{v+1}, and\ v \le \lceil \log_2(q - p + 2)\rceil.$$

### D.4 Technical Lemmas

In this section, we will present several technical lemmas used in our proof.

**Lemma 12** (Lemma 3.1 of Srebro et al. [2010]). *For an $L$-smooth and nonnegative function $f : \mathcal{X} \mapsto \mathbb{R}_+$,*

$$\|\nabla f(\mathbf{x})\|_2 \le \sqrt{4Lf(\mathbf{x})},\ \forall \mathbf{x} \in \mathcal{X}.$$

**Lemma 13** (Lemma 3.5 of Auer et al. [2002]). *Let $l_1, \ldots, l_T$ be non-negative real numbers. Then:*

$$\sum_{t=1}^{T} \frac{l_t}{\sqrt{\delta + \sum_{i=1}^{t} l_i}} \le 2\left(\sqrt{\delta + \sum_{t=1}^{T} l_t} - \sqrt{\delta}\right).$$

**Lemma 14** (Lemma 14 of Gaillard et al. [2014]). *Let $a_0 > 0$ and $a_1, \ldots, a_m \in [0, 1]$ be real numbers and let $f : (0, +\infty) \mapsto [0, +\infty)$ be a non-increasing function. Then*

$$\sum_{i=1}^{m} a_i f(a_0 + \cdots + a_{i-1}) \le f(a_0) + \int_{a_0}^{a_0 + a_1 + \cdots + a_m} f(u)du.$$

**Lemma 15** (Lemma 4.8 of Pogodin and Lattimore [2019]). *Let $a_1, a_2, \ldots, a_T$*

$$\sum_{t=1}^{T} \frac{a_t}{\sqrt{1 + \sum_{s=1}^{t-1} a_s}} \le 4\sqrt{1 + \sum_{t=1}^{T} a_t} + \max_{t \in [T]} a_t.$$

**Lemma 16** (Lemma 5 of Shalev-Shwartz [2007]). *For any $x, y, a \in \mathbb{R}_+$ that satisfies $x - y \le \sqrt{ax}$,*

$$x - y \le \sqrt{ay} + a.$$

Based on Lemma 16, we can achieve the following result.

**Lemma 17.** *For any $x, y, a, b \in \mathbb{R}_+$ that satisfies $x - y \le \sqrt{ax} + b$,*

$$x - y \le \sqrt{ay + ab} + a + b.$$

**Lemma 18** (Lemma 13 of Gaillard et al. [2014]). *For all $x > 0$ and all $\alpha \ge 1$, we have*

$$x \le x^\alpha + \frac{\alpha - 1}{e}.$$