# OpenReview forum: "Efficient Methods for Non-stationary Online Learning"
_NeurIPS.cc/2022/Conference — NeurIPS 2022 Accept_

### Official Review · Reviewer_JRs5 · 2022-06-29

**Rating:** 8
**Confidence:** 3
**Soundness:** 3 good
**Presentation:** 4 excellent
**Contribution:** 3 good

**Summary:**

This work tackles the topic of time complexity in modern online algorithms. Indeed, to deal with dynamic environments, modern algorithms use a 2-layers structure to obtain strong predictions and it implies at each time step $O(\log(T)$ projections over a possibly complex convex set. Those projections may be time-expensive in practice, and not suitable for many real-life situations.
That is why authors propose a generic mechanism which only involves $O(1)$ projections at each time step. They apply their mechanism onto three distinct algorithms dedicated to minimise different types of regrets: dynamic regret, adaptive regret and interval dynamic regret.
Authors also propose in Appendix A a toy experiment illustrating the efficiency of their procedures.

**Questions:**

- I am wondering about the lifetime of the base learners in Alg.2. In lines 8-10, how do we choose the ending points of a new base learner? Is it arbitrary?
- I am not sure to understand what have been the motivation to design $\Tilde{\mathcal{C}}$ in Eq. 10)? Is it possible to have more explanations?
- It is said that the code is available but I did not find any python file in the supplementary nor a link to an anonymous Github, did I miss something?
- An open question: authors state in their conclusion that it would be challenging to incorporate optimistic online learning within this work. But could not we incorporate, at each time, the optimistic information directly within the base algorithm and see where it goes?

**Strengths And Weaknesses:**

Strengths:

- A pedagogical work, clear and well-written.
- The goals of the paper are well stated. The theoretical results are, for me, impacting and come with a convincing toy experiment.
- Any new result or algorithm comes with welcomed explanations and discussions.


Weaknesses:
- It would have been good to mention more the existence of experiments within the main document. I believe they constitute an important point of this work but they seems to be quite hidden.
- I think it would have been good to mention a few learning problems which involves concrete complicated convex sets on which the projection step is costful.

Conclusion:

I congratulate the authors for their work. To me,  they performed a significant step in online learning as they reduced time complexity of modern algorithms with the use of insightful surrogates without sacrificing any performance guarantee. The proposed experiment numerically attest this fact. Furthermore, the proposed work is not only impacting but also clear and well-written, it has been a pleasure to read this document.
Finally I precise that I am not that familiar with the literature around the adaptive regret, thus I may overestimate the strength of some proposed results but I am confident in the fact that Alg.1, which is a twisted improved Ader is impacting in the dynamic regret field.
However, for all those positive reasons, I give this paper a 8.




Remark:
- As I am not familiar with Adaptive Regret literature, I only checked the proofs for Dynamic Regret which are for me, correct.

---

> ### Author Response · Authors · 2022-08-01
> **Thanks for your appreciation and great questions! (1/2)**
>
> Thanks for your appreciation and careful reviews. We answer your main questions as follows. The questions are mainly due to the unclear presentation, and we will surely improve the writing to make them clear in the next version.
>
> ------
> **Q1**. "I am wondering about the lifetime of the base learners in Alg.2. In lines 8-10, how do we choose the ending points of a new base learner? Is it arbitrary?"
>
> **A1**. We are sorry for the inexact writing that leads to confusion.  We clarify that the choice of the ending point of a new base-learner is not arbitrary, and it is specified by our construction of geometric covers and markers. Below, we explain these pseudocodes.
>
> In Line 8 of Algorithm 2, we say "remove base-learner whose ending point $e_k = m+1$", where the "ending point" is specified by the Compact Geometric Cover (CGC) $\mathcal{C}$ defined in Line 10. It can be verified that for each natural number $a \in \mathbb{N}$, there exists and only exists one corresponding interval [a,b] in CGC $\mathcal{C}$ defined in Line 10 of Algorithm 2. So whenever the cumulative loss exceeds the threshold such that we register the value $m$, there always exists an interval $[m, j-1]$ in the CGC $\mathcal{C}$ (noted in Line 10).
>
> We clarify that "ending point" is something defined in the level of the index of markers and is **not the ending time stamp** that denotes the exact time stamp that the base-learner stops the updates. Indeed, to achieve a small-loss bound, one needs to adaptively decide the initialization of a new base-learner, and the time stamp when the cumulative loss exceeds a certain threshold is called the "marker". As a result, those starting/ending time stamps cannot be known in advance, as they are determined according to the cumulative loss on the fly. In the paper, we introduce the notation $\tilde{\mathcal{C}}$ in Eq. (10) to denote the set of exact starting/end time stamps of the base-learners, but those quantities cannot be known in advance for the algorithm implementations. So a more precise statement for Line 8 of Algorithm 2 would be "remove base-learners $B_k$ whose ending point $e_k = m+1$ (and then its ending time stamp is $s_{e_k} = s_{m+1}$)". We will revise the paper to add more explanations. Thanks!
>
> ----
> **Q2**. "I am not sure to understand what have been the motivation to design in Eq. (10)? Is it possible to have more explanations?"
>
> **A2**. Thanks for your comments. Yes, we should explain more about Problem-dependent Geometric Covers (PGC), and this will be added in the revised version.
>
> To achieve problem-dependent bounds, we need to adaptively decide the initialization of a new base-learner, and the time stamp when the cumulative loss exceeds a certain threshold is called the "marker". Eq. (10) denotes our constructed PGC, which bridges the "marker" language and the exact starting time and ending time for each base-learner. This is very important in the analysis. The reason that $\tilde{\mathcal{C}}$ seemingly didn't appear in the algorithm is because *those starting/ending time stamps cannot be known in advance*, as they are determined according to the cumulative loss on the fly. But to avoid confusion and to be more precise, in Algorithm 2 Line 10, we can state more concretely as "Initialize a new base-learner with ending point $e_m = j$ satisfying $[m,j-1] \in \mathcal{C}$ and then, the ending *time stamp* of this base-learner is the marker $s_{e_m}$ appeared in the PGC $\tilde{\mathcal{C}}$". A similar modification can be made in Line 8. We will add more explanations to make this clear. Thanks!
>
> ------
> **Q3**. "It is said that the code is available but I did not find any python file in the supplementary nor a link to an anonymous Github, did I miss something?"
>
> **A3**. Thank you for the reminder. We have uploaded our code to an anonymous link, https://anonymous.4open.science/r/anonymized-conf-code-PE.

---

> > ### Author Response · Authors · 2022-08-01
> > **Thanks for your appreciation and great questions! (2/2)**
> >
> > **Q4**. "An open question: authors state in their conclusion that it would be challenging to incorporate optimistic online learning within this work. But could not we incorporate, at each time, the optimistic information directly within the base algorithm and see where it goes?"
> >
> > **A4**. Thank you for this in-depth question. Incorporating optimism looks easy at an initial thought, but it is, unfortunately, nontrivial to achieve (please correct me if I am wrong, which I will be definitely happy to see). The reason is due to the surrogate loss technique used in the reduction scheme. Specifically, given an optimism $M_t$ that approximates the true gradient $\nabla f_t(x_t)$, we need to come up with a "surrogate optimism" $\tilde{M}_t$ to approximate the gradient of the surrogate loss $\nabla g_t(y_t)$, in which we hope that $\Vert\tilde{M}_t - \nabla g_t(y_t)\Vert$ can be upper bounded by $\Vert M_t - \nabla f_t(x_t)\Vert$. But such a construction of $\tilde{M}_t$ is not easy to attain because the natural construction of $\tilde{M}_t$ will depend on $x_t$, while $x_t$ also depends on $\tilde{M}_t$ (recall the update step in optimistic online learning). So one needs to solve an equation to derive an appropriate $\tilde{M}_t$, which is actually non-trivial, especially given the fact that $x_t$ is in a meta-base aggregation form and also requires to be projected back to the constrained feasible set.
> >
> > We hope the above discussions will clarify the question. But this is definitely an important problem to consider, and we take it as future work.

---

> > > ### Comment · Reviewer_JRs5 · 2022-08-08
> > > **Thank you for your answers**
> > >
> > > I thank the authors for their additional explanations and their additions in the revised version.
> > >
> > > I remain utterly convinced by this contribution.

---

### Official Review · Reviewer_9mtH · 2022-07-11

**Rating:** 7
**Confidence:** 3
**Soundness:** 4 excellent
**Presentation:** 4 excellent
**Contribution:** 3 good

**Summary:**

The paper presents a reduction that achieves 1 projection complexity for algorithms with low dynamic regret and adaptive regret, which typically have $O(\log T)$ projection complexity, without degrading their regret guarantees. Since the projection operation can be a major bottleneck when the decision set is complicated, the reduction has significant computational advantages over the original algorithm. In cases where the reduction cannot be straightforwardly applied, the work specifies modifications under which 1 projection complexity can be achieved.


**Questions:**

To make the impact clear, it would be helpful to explicitly state some settings/problems where the projection step is expensive in the main paper.


**Limitations:**

Yes.

**Strengths And Weaknesses:**

Strengths
- The paper proposes a method to address the computational efficiency problem in dynamic regret/adaptive regret minimization, which has applications in many domains. The proposed method achieves significant computational savings when projecting onto the convex decision set is expensive.
- The reduction is novel as far as I know, and serves as the backbone of a series of algorithms that achieve 1 projection complexity without worsening regret guarantees. When the original algorithm doesn’t satisfy the conditions of the reduction, nontrivial changes are presented to accommodate the requirements, along with their theoretical justifications.
- The paper is very clearly written and concisely presented.

Weaknesses
- See questions.

---

> ### Author Response · Authors · 2022-08-01
> **Thanks for your appreciation and suggestions!**
>
> Thanks for your appreciation and helpful comments. Given that an additional page is allowed in the final camera-ready version, we are willing to add more specific problems with high projection complexity, for example, online principal component analysis, some online decision-making problems that can be reduced to online convex optimization and thus require projecting onto the complicated feasible domain, etc. We will add those in the revised version to improve the presentation. Thanks!

---

### Official Review · Reviewer_2iQY · 2022-07-11

**Rating:** 6
**Confidence:** 3
**Soundness:** 3 good
**Presentation:** 3 good
**Contribution:** 3 good

**Summary:**

This paper studies methods making online learning with non-stationary comparators more efficient in terms of the number of projections onto the feasible set in the OCO setting. The authors begin with a typical meta-algorithm for the dynamic/adaptive regret setting, where the meta-algorithm averages predictions over many base algorithms, but each base algorithm (of which there are typically O(log(T)) requires a projection step. The main idea is to replace the feasible set with the smallest Euclidean ball containing it (making the projection trivial) and the loss function with a surrogate loss that allows for the same regret guarantee, run the base algorithms with this easy projection, then do the projection onto the action set afterwards. Since  single projection can be shared by all the base algorithms, resulting in a log(T) reduction in the number of projections (assuming projections to Euclidean balls are free).


This reduction is applied to different algorithms to build projection-efficient algorithms for dynamic regret and small-loss dynamic regret. Finally, the authors also present a new algorithm for interval dynamic regret, which is nice; the SOTA algorithm for this setting, SACS, needs to be heavily modified to obtain the small-loss bound.


**Questions:**

The authors make a compelling case for the applicability of their reduction technique and derive several algorithms with great computational properties. Three are presented in the paper. Even without the goal of reducing the number of projections, the surrogate function reduction seems to be a powerful technique and the authors give good reason to know about it. The range of results obtained are impressive.

There are some small presentation issues. It becomes difficult to keep track of all the meta or base algorithm substitutions, some some assistance (e.g. in the form of a table) would be appreciated.

I would also appreciate more discussion about the proof techniques, especially regarding what is novel. One of the main takeaways of this work is the utility of the surrogate loss, and some proof outlines would help.

A few other small presentation issues:
Use "twists" a bit too much, especially in the intro.
Small grammatical mistakes, i.e.
166: the feasibility -> feasibility
174 (and other places): 1 projection -> one projection
Since empirical experiments aren't in the main body, I would not reference them in the abstract.


**Limitations:**

I think a more thorough discussion of the algorithmic trade-offs needed to ensure single projections is necessary; see my earlier comments. No societal impact discussion is presented or expected.

**Strengths And Weaknesses:**

The authors make a compelling case for the applicability of their reduction technique and derive several algorithms with great computational properties. Three are presented in the paper. Even without the goal of reducing the number of projections, the surrogate function reduction seems to be a powerful technique and the authors give good reason to know about it. The range of results obtained are impressive.

There are some small presentation issues. It becomes difficult to keep track of all the meta or base algorithm substitutions, some some assistance (e.g. in the form of a table) would be appreciated.

I would also appreciate more discussion about the proof techniques, especially regarding what is novel. One of the main takeaways of this work is the utility of the surrogate loss, and some proof outlines would help.

A few other small presentation issues:
Use "twists" a bit too much, especially in the intro.
Small grammatical mistakes, i.e.
166: the feasibility -> feasibility
174 (and other places): 1 projection -> one projection
Since empirical experiments aren't in the main body, I would not reference them in the abstract.

---

> ### Author Response · Authors · 2022-08-01
> **Thanks for your appreciation and suggestions!**
>
> Thanks for your appreciation and insightful review. Below we address your questions.
>
> -------
> **Q1**. "I would also appreciate more discussion about the proof techniques, especially regarding what is novel. One of the main takeaways of this work is the utility of the surrogate loss, and some proof outlines would help."
>
> **A1**. As mentioned in Section 1.3 of the main text, one of the major backbones of the reduction scheme is the surrogate loss technique, while in order to fit the reduction scheme, one needs to modify existing methods from various aspects, including the meta-algorithm (using Adapt-ML-Prod to achieve a second-order regret bound), the geometric covers (using threshold generating function to determine the covers adaptively), etc. Thanks for the comment. We will add more discussions on the technical novelty in the paper and also provide a summary of proof outlines to help readers better understand the role of those techniques in the proofs.
>
> -------
> **Q2**. other presentations issues (some assistance for algorithms, wording, etc)
>
> **A2**. Thanks, we will improve the presentation according to your suggestions.
>
> -------
> **Q3**. "I think a more thorough discussion of the algorithmic trade-offs needed to ensure single projections is necessary; see my earlier comments."
>
> **A3**. Thank you for the suggestion. We will add more discussion on this point. Briefly speaking, due to the specific construction of the surrogate loss, we require the online algorithm to query function value and gradient value only once per iteration. Wrapping the reduction scheme, the algorithm can then enjoy one projection onto the original feasible domain per round. We will emphasize this algorithmic trade-off to ensure a single projection in the revised version.

---

> > ### Comment · Reviewer_2iQY · 2022-08-09
> > **Sounds good to me.**
> >
> > I continue to think this is a good paper we should accept.

---

### Official Review · Reviewer_zt1M · 2022-07-12

**Rating:** 7
**Confidence:** 4
**Soundness:** 4 excellent
**Presentation:** 4 excellent
**Contribution:** 3 good

**Summary:**

The paper presents a generic reduction-based method for obtaining online learning algorithms with adaptive and/or dynamic regret guarantees that have low “projection complexity”, i.e., use the projection operation as few times per round as possible. In fact, subject to the condition that the reduced algorithm queries the gradient and evaluates the loss function at most once per round, it is demonstrated that this reduction results in the algorithm doing at most one projection per round. The relevance of this result is that the dynamic/adaptive regret literature has recently increasingly adopted a two-level hyper-expert setup where decisions of a substantial number of hyper-experts are aggregated every round, and such methods naively would do a projection per expert per round --- such projections being a computationally costly operation. In the paper, the approach is illustrated by applying it to several algorithms (Ader, SACS, AOA) with state-of-the-art dynamic/adaptive/interval dynamic regret bounds --- thus reducing these algorithms’ projection complexity while keeping their performance guarantees.

**Questions:**

I am overall happy with the paper’s presentation. As one suggestion, I would recommend --- for the purposes of further driving home the main message (that low projection complexity, as achieved through the proposed reduction, does substantial good for the runtime not just in theory, but also in practice) --- that the authors try to find a way to move some improved runtime plots from the appendix into the main part of the paper.

**Strengths And Weaknesses:**

Overall, I consider this paper to be a high-quality submission: On the one hand, it thoroughly develops and applies, on state-of-the-art terrain, a framework for reducing the computational complexity of OCO algorithms. On the other hand, it exhibits high-quality writing/presentation --- providing a good, largely self-contained, overview of important related literature and results, and giving insight into the developed reduction methods and how to apply them in “practice”.

In terms of originality, the backbone of the proposed reduction method (“replace the original domain by a Euclidean ball with an appropriately defined surrogate loss, and project back onto original domain”) to a substantial degree originates in a line of work by Cutkosky and Orabona on black-box reductions for parameter-free online learning. The main contribution of the current paper is in recognizing the generic potential of this type of reduction to generate low-projection-complexity state-of-the-art dynamic/adaptive regret algorithms, and showing how to get through the weeds of some complex recent algorithms like SACS in order to bring them into the appropriate shape for this reduction to result in one-per-round projection complexity (which essentially amounts to modifying these algorithms so that they have one gradient/function evaluation per round). I believe, on balance, that this is a high-quality and sufficiently original contribution to the online learning literature.

---

> ### Author Response · Authors · 2022-08-01
> **Thanks for your appreciation!**
>
> Many thanks for your careful review and constructive comments. In the next version, we will try to move some empirical results into the main text. We believe this is very feasible, given that an additional page is allowed in the camera-ready version. Thanks!

---

> > ### Comment · Reviewer_zt1M · 2022-08-09
> > **Thank you for the responses and the contribution!**
> >
> > Thank you for addressing the reviewers' issues --- and thank you to the other reviewers for their comments.
> >
> > This paper both promises impact on the community, and is very clearly written. I will continue to recommend acceptance.

---

### Meta-Review · Area_Chair_ak2y · 2022-08-24

**Recommendation:** Accept
**Confidence:** Certain

**Metareview:**

The authors have successfully developed a powerful surrogate-based reduction framework for greatly reducing the projection complexity of online learning algorithms with low adaptive and dynamic regret while preserving regret bounds. They further demonstrate how to use this approach for several recent algorithms with the best-known regret bounds. An additional result is the first algorithm that enjoys a small-loss type bound for the interval dynamic regret (for convex, smooth functions), and the algorithm achieving this also requires only one projection onto the feasible set per round. The reviewers are unanimous in the high quality of this work. This is an impressive and welcome contribution to the proceedings. Congratulations on your fine work!

**Award:**

No

---

### Decision · Program_Chairs · 2022-09-14

Accept